# The M-phase regulatory phosphatase PP2A-B55δ opposes protein kinase A on Arpp19 to initiate meiotic division

Tom Lemonnier[1,3], Enrico Maria Daldello [1,3], Robert Poulhe[1], Tran Le[1], Marika Miot[1], Laurent Lignières [2], Catherine Jessus[1] & Aude Dupré [1✉]

Oocytes are held in meiotic prophase for prolonged periods until hormonal signals trigger meiotic divisions. Key players of M-phase entry are the opposing Cdk1 kinase and PP2A-B55δ phosphatase. In *Xenopus*, the protein Arpp19, phosphorylated at serine 67 by Greatwall, plays an essential role in inhibiting PP2A-B55δ, promoting Cdk1 activation. Furthermore, Arpp19 has an earlier role in maintaining the prophase arrest through a second serine (S109) phosphorylated by PKA. Prophase release, induced by progesterone, relies on Arpp19 dephosphorylation at S109, owing to an unknown phosphatase. Here, we identified this phosphatase as PP2A-B55δ. In prophase, PKA and PP2A-B55δ are simultaneously active, suggesting the presence of other important targets for both enzymes. The drop in PKA activity induced by progesterone enables PP2A-B55δ to dephosphorylate S109, unlocking the prophase block. Hence, PP2A-B55δ acts critically on Arpp19 on two distinct sites, opposing PKA and Greatwall to orchestrate the prophase release and M-phase entry.

[1] Sorbonne Université, CNRS, Laboratoire de Biologie du Développement—Institut de Biologie Paris Seine, LBD—IBPS, Paris, France. [2] Université de Paris, CNRS, Institut Jacques Monod, Paris, France. [3] These authors contributed equally: Tom Lemonnier, Enrico Maria Daldello. ✉email: aude-isabelle.dupre@upmc.fr

In vertebrates, oocyte meiotic divisions are controlled by the cAMP-dependent protein kinase A, PKA. The activity of PKA is responsible for arresting fully-grown oocytes in prophase of the first meiotic division for long-lasting periods in order to allow cell growth and nutrient accumulation required for fertilization and the early embryogenesis[1,2]. The post-transcriptional mechanisms controlled by PKA in oocytes were largely unknown until the identification of the protein Arpp19 as its long sought substrate in *Xenopus*[3]. In this species, Arpp19 phosphorylation at S109 by PKA is critical to maintain oocytes arrested in prophase. Release from the prophase block is promoted by progesterone, which triggers within 0.5 h a drop in the cAMP concentration and the down-regulation of PKA activity. As a result, Arpp19 is dephosphorylated at S109 and unlocks a signaling pathway that leads within 3 to 5 h to the activation of the Cdk1-Cyclin B complex and Greatwall kinase (Gwl), or MPF (M-phase promoting factor), the universal inducer of eukaryotic cell division[4]. Once activated, Cdk1-Cyclin B complexes trigger the resumption of the first meiotic cell division which starts with nuclear envelope breakdown, termed germinal vesicle breakdown (GVBD) in oocytes[5].

The biochemical steps linking PKA downregulation and Arpp19 dephosphorylation at S109 to Cdk1-Cyclin B activation involve the translation of two proteins, Cyclin B and the kinase Mos[6]. Newly synthesized Cyclin B associates with free Cdk1 while Mos indirectly activates MAPK (mitogen-activated protein kinase). Both events allow the formation of newly active Cdk1-Cyclin B complexes and the hyperphosphorylation of the two opposed regulators of Cdk1, the phosphatase Cdc25 and the kinase Myt1[5]. Upon phosphorylation, Cdc25 activates Cdk1 by dephosphorylating T14 and Y15, whereas its antagonistic enzyme, Myt1, is concomitantly inhibited[7]. Importantly, Cdk1 activation also requires the inhibition of a specific phosphatase, the PP2A-B55δ isoform, which counteracts Cdk1-dependent phosphorylations of mitotic/meiotic substrates, likely including Cdc25 and Myt1[8-12]. PP2A-B55δ inhibition is achieved by Arpp19, a specific inhibitor of this phosphatase when phosphorylated at S67 by Gwl[13,14]. Hence, Arpp19 is a central player of meiosis resumption through two distinct functions at two different periods. In prophase oocytes, Arpp19 is phosphorylated by PKA at S109 and holds the prophase block. Upon hormonal stimulation, its dephosphorylation at S109 occurs within 1 h and unlocks a signaling pathway leading to meiotic division. At the very end of this molecular pathway, 3–5 h later, Arpp19 is phosphorylated at S67 by Gwl and inactivates PP2A-B55δ, hence promoting Cdk1-Cyclin B activation[5].

The molecular regulation of this last step, involving Gwl, Arpp19 and PP2A, has been well deciphered in mitosis and meiosis. PP2A is a major S/T protein phosphatase conserved across eukaryotes. It acts as a heterotrimeric complex composed of a catalytic C subunit, a scaffolding A subunit, and a variable B subunit, which regulates PP2A intracellular localization and substrate specificity[15]. Eukaryotes have four B-subunit families known as B (B55), B' (B56), B'' (PR72) and B''' (striatin), each of them comprising several very close isoforms whose number differs according to the species[16]. From yeast to humans, Gwl kinase displays conserved functions in mitosis[17], which are mediated by the phosphorylation of its sole substrate identified so far, Arpp19, and its paralog, α-endosulfine (ENSA). ENSA and Arpp19 derive from the duplication of an ancestral gene that evolved through new duplications and losses (http://www.ensembl.org; http://www.treefam.org), providing from one to four homologs in the different species[18,19]. Both Arpp19 and ENSA share highly conserved sequence identity/similarity across most eukaryotes. Their most conserved region is centered around the FDS$_{67}$GDY motif (*X. laevis* numbering). Within this motif, S67 is phosphorylated

by Gwl to the same extent in Arpp19 and ENSA, generating phosphorylated proteins that bind to and equally inhibit the specific PP2A-B55δ isoform by titrating the phosphatase away from all other substrates and making themselves its preferential substrates[20,21]. Whether Arpp19 and ENSA display specific functions is not clear, although some evidence shows that, unlike ENSA, Arpp19 plays an essential role during mouse embryogenesis and in regulating mitotic and meiotic divisions[22]. In *Xenopus* oocyte, it is clearly established that S67 phosphorylation of Arpp19 by Gwl promotes its binding to PP2A-B55δ and the inhibition of the phosphatase[23,24]. Released from the activity of its opposite enzyme, Cdk1 phosphorylates its two antagonistic regulators, Cdc25 and Myt1, setting up the positive feedback loop responsible for its abrupt and full activation[5]. Importantly, the activation of the Gwl/Arpp19/PP2A-B55δ module depends on Cdk1 activity[24-27], positioning this module inside the auto-activation loop. Hence, the antagonistic relationship between Arpp19-Gwl and PP2A-B55δ greatly contributes to the abruptness and irreversibility of cell division entry[28].

PKA phosphorylates ENSA and Arpp19 at a consensus RKP/SS$_{109}$LV motif (*X. laevis* numbering) conserved among most animals. Specific functions have been attributed to the PKA-phosphorylated form of Arpp19/ENSA, notably in striatal neurons upon dopaminergic stimulation[29]. No specific role related to cell division had been described until we discovered that Arpp19 phosphorylation by PKA is essential to arrest *Xenopus* oocytes in prophase[3]. The S109 phosphorylation by PKA does not impede the phosphorylation at S67 by Gwl nor its ability to inhibit PP2A-B55δ when phosphorylated at S67[26]. Moreover, Arpp19 is rephosphorylated at S109 by an unknown kinase distinct from PKA, concomitantly with its S67 phosphorylation by Gwl, at time of Cdk1 activation[3]. Thus, the events controlled by the S109 phosphorylation of Arpp19 that maintain the prophase block in oocytes remain an open question. Another key issue to unravel the prophase release regards the identity of the phosphatase that dephosphorylates Arpp19 at S109 at the onset of meiosis resumption. Since this event is important to unlock the transduction pathway leading to cell division, this unidentified phosphatase is a critical player of oocyte meiotic division.

Here, we identify PP2A-B55δ as the phosphatase that dephosphorylates Arpp19 at S109, thus enabling oocytes to resume meiosis. The level of Arpp19 phosphorylated at S109 in prophase-arrested oocytes results from a balance between PKA and PP2A-B55δ activities in favor of the kinase. Upon hormonal stimulation, PP2A-B55δ activity remains unchanged while PKA is downregulated, leading to the partial dephosphorylation of Arpp19 at S109 that unlocks the prophase arrest. Therefore, the timing of meiosis resumption relies on the temporal coordination of S109 and S67 phosphorylations of Arpp19, orchestrated by one single phosphatase, PP2A-B55δ, opposing two kinases, PKA and Gwl.

## Results

**Active Arpp19 dephosphorylation at S109 opposed by PKA in prophase oocytes.** The S109 residue of Arpp19 phosphorylated by PKA in prophase oocytes is dephosphorylated in response to progesterone by an unknown phosphatase[3], termed S109-phosphatase until its identification. The level of S109-phosphorylated Arpp19 in prophase-arrested oocytes could result from either the sole activity of PKA or a balance between PKA and S109-phosphatase in favor of PKA. To address this issue, we first assayed S109-phosphatase activity in extracts from prophase oocytes. As a substrate, we used GST-tagged Arpp19 previously in vitro phosphorylated at S109 by PKA (pS109-GST-Arpp19)[26]. Note that GST-Arpp19 is partially proteolyzed during either its expression in bacteria or its purification,

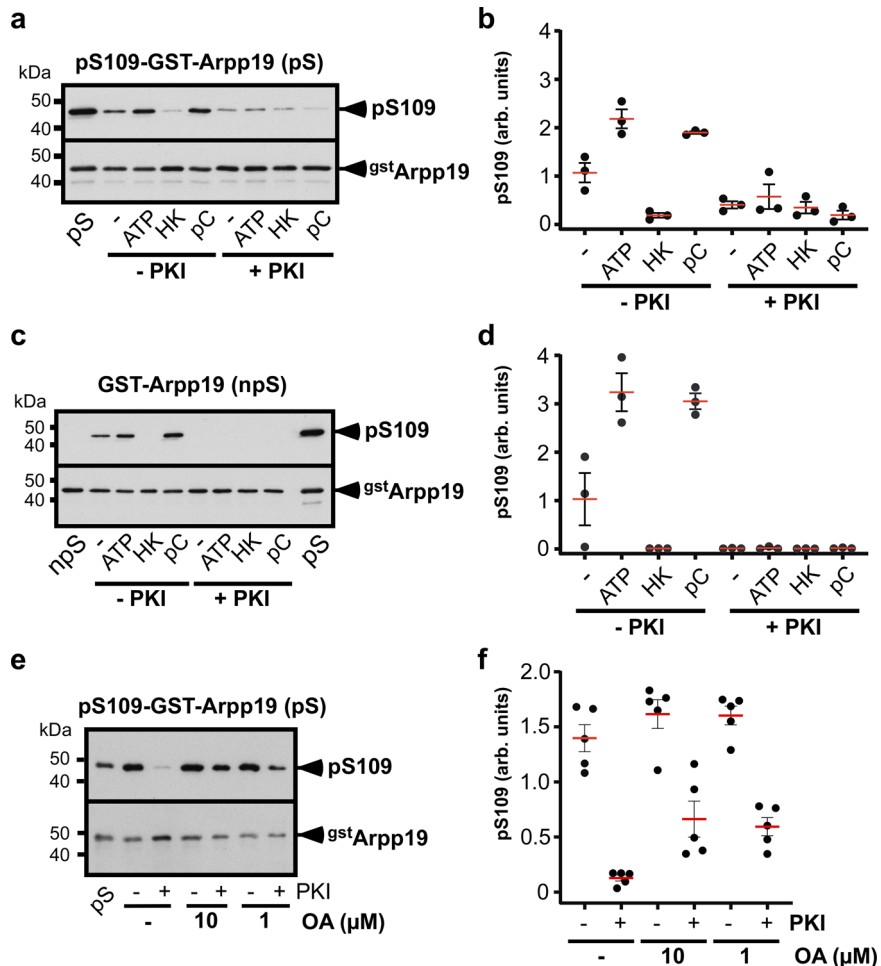

**Fig. 1 S109-phosphatase is active, OA-sensitive and counterbalanced by PKA in prophase extracts. a–d** Extracts from prophase oocytes were supplemented or not with either ATP or hexokinase and glucose (HK), in the presence or not of phosphocreatine (pC). Extracts were then incubated or not with PKI and either pS109-GST-Arpp19 (pS: phosphorylated substrate) (**a–b**) or GST-Arpp19 (npS: non-phosphorylated substrate) (**c–d**). S109 phosphorylation of GST-Arpp19 (pS109) and total GST-Arpp19 (gstArpp19) were analyzed by western blot with phospho-S109-Arpp19 and GST antibodies. S109-phosphatase activity: one representative experiment (**a**) and quantifications of S109 phosphorylation from 3 independent experiments (**b**). S109-kinase activity: one representative experiment (**c**) and quantifications of S109 phosphorylation from 3 independent experiments (**d**). **e–f** Prophase extracts were incubated or not with 1 μM or 10 μM okadaic acid (OA), supplemented or not with PKI and further incubated with pS109-GST-Arpp19 (pS) in the presence of ATP. S109 phosphorylation of GST-Arpp19 (pS109) and total GST-Arpp19 (gstArpp19) were analyzed as in panel (**a**). One representative experiment and quantifications of S109 phosphorylation from 5 independent experiments are presented in (**e**) and (**f**) respectively. For quantifications, data are shown as mean (red bars) ± SEM. Each dot represents one experiment. arb. units: arbitrary units. kDa: kiloDalton. Source data are provided as a Source Data file.

occasionally producing a band of lower molecular weight than the full-length protein that lacks S109 but is recognized by the anti-GST antibody (Supplementary Fig. 1). pS109-GST-Arpp19 was coupled to GSH-beads and then incubated in prophase extracts. S109 phosphorylation of pS109-GST-Arpp19 recovered from extracts was monitored by western blot using a specific phospho-S109-Arpp19 antibody[3]. Arpp19 was efficiently dephosphorylated at S109 (Fig. 1a and b), showing that S109-phosphatase is active in prophase extracts. Oocyte lysis leads to ATP hydrolysis and as a result, oocyte extracts contain low levels of ATP that prevent kinases from functioning. Interestingly, adding ATP reduced Arpp19 dephosphorylation at S109 (Fig. 1a and b). To control the ATP amount, prophase extracts were supplemented with hexokinase, which fully depletes ATP[30]. Under this condition, Arpp19 was strongly dephosphorylated at S109 (Fig. 1a and b). In contrast, in the presence of phosphocreatine that replenishes ATP[30], Arpp19 dephosphorylation at S109 was severely impaired (Fig. 1a and b). Altogether, these results suggest that a kinase

counteracts S109-phosphatase. When the specific inhibitor of PKA, PKI[31], was added to extracts in the presence of ATP, pS109-GST-Arpp19 was efficiently dephosphorylated (Fig. 1a and b). This indicates that S109-phosphatase activity is counterbalanced by PKA in prophase extracts. Furthermore, non-phosphorylated GST-Arpp19 was efficiently phosphorylated at S109 in extracts, a process that was dependent on ATP and sensitive to PKI (Fig. 1c and d). These results demonstrate that PKA and S109-phosphatase are both active and regulate Arpp19 phosphorylation at S109 in prophase extracts, with PKA dominating. In subsequent experiments, PKI was included in order to block PKA activity and to facilitate analysis of its opposing phosphatase.

We next determined whether S109-phosphatase belongs to the group of okadaic acid (OA)-sensitive phosphatases, which includes PP1, PP2A and phosphatases related to PP2A[16,32]. Prophase extracts were first supplemented with 1 or 10 μM OA. PKI and the S109 phosphatase substrate, pS109-GST-Arpp19, were then added in the presence of ATP. Both OA concentrations

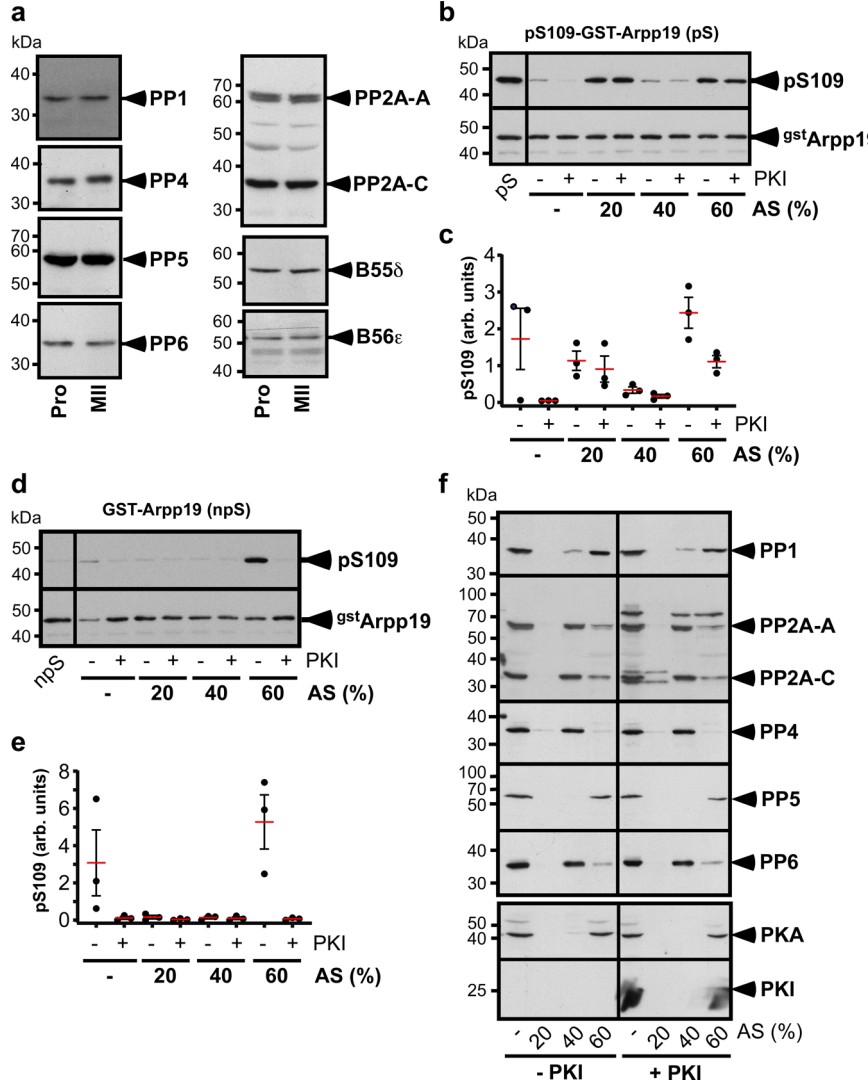

**Fig. 2 Ammonium sulfate precipitation separates S109-phosphatase from PKA, PP1 and PP5. a** Western blot analysis of various S/T phosphatases sensitive to OA in lysates from prophase (Pro) or metaphase II (MII) oocytes using specific antibodies directed against catalytic subunits of PP1, PP2A (PP2A-C), PP4, PP5, PP6, and PP2A-regulatory subunit A (PP2A-A), B55δδ, and B56ε. The experiment was repeated 3 times with similar results. **b**–**e** Prophase extracts supplemented or not with PKI were precipitated by serial addition of ammonium sulfate (AS) as indicated. (–): Starting extracts without AS. Pellets were recovered and used for enzymatic assays and western blots with phospho-S109-Arpp19 and GST antibodies. S109-phosphatase activity was assayed using pS109-GST-Arpp19 (pS: phosphorylated substrate): one representative experiment (**b**) and quantifications of S109 phosphorylation from 3 independent experiments (**c**). **d**–**e** PKA activity was assayed using GST-Arpp19 (npS: non-phosphorylated substrate): one representative experiment (**d**) and quantifications of S109 phosphorylation from 3 independent experiments (**e**). For quantifications, data are presented as mean (red bars) ± SEM. Each dot represents one experiment. arb. units: arbitrary units. **f** Western blot analysis of initial extracts (–) and AS precipitates using specific antibodies directed against catalytic subunits of PP1, PP2A (PP2A-C), PP4, PP5, PP6, PKA, and against PP2A scaffold subunit A (PP2A-A) and PKI. The experiment was repeated 3 times with similar results. kDa: kiloDalton. Source data are provided as a Source Data file.

prevented Arpp19 dephosphorylation at S109 (Fig. 1e and f). Hence, S109-phosphatase is an OA-sensitive phosphatase that is active and antagonizes PKA in prophase extracts.

**S109-phosphatase is distinct from PP1 and PP5**. As the S109-phosphatase is inhibited by OA, we analyzed which ones of the known OA-sensitive phosphatases are expressed in *Xenopus* oocytes. Western blots revealed that the catalytic subunits of PP1, PP4, PP5, PP6 and PP2A (PP2A-C) as well as PP2A structural (PP2A-A) and regulatory (B55δ and B56ε) subunits are expressed at the same level in prophase- and metaphase II-arrested oocytes (Fig. 2a). In an attempt to identify which one corresponds to S109-phosphatase, prophase extracts were fractionated by

precipitation using increasing amounts of ammonium sulfate. S109-phosphatase and PKA activities were then assayed, using as substrates pS109-GST-Arpp19 and GST-Arpp19 respectively. S109-phosphatase activity was recovered in the 40% precipitate (Fig. 2b and c) whereas PKA activity was precipitated by 60% ammonium sulfate (Fig. 2d and e). S109-phosphatase activity did no longer depend on PKI in the 40% precipitate (Fig. 2b and c), in agreement with the absence of PKA activity in this precipitate (Fig. 2d and e).

Each precipitated fraction was further analyzed by western blot with antibodies against OA-sensitive phosphatases, PKA and PKI. Consistently with the S109 phosphorylation activity measured in Fig. 2d and e, PKA was recovered in the 60% ammonium sulfate precipitate (Fig. 2f). PP2A, PP4 and PP6, as well as the scaffold

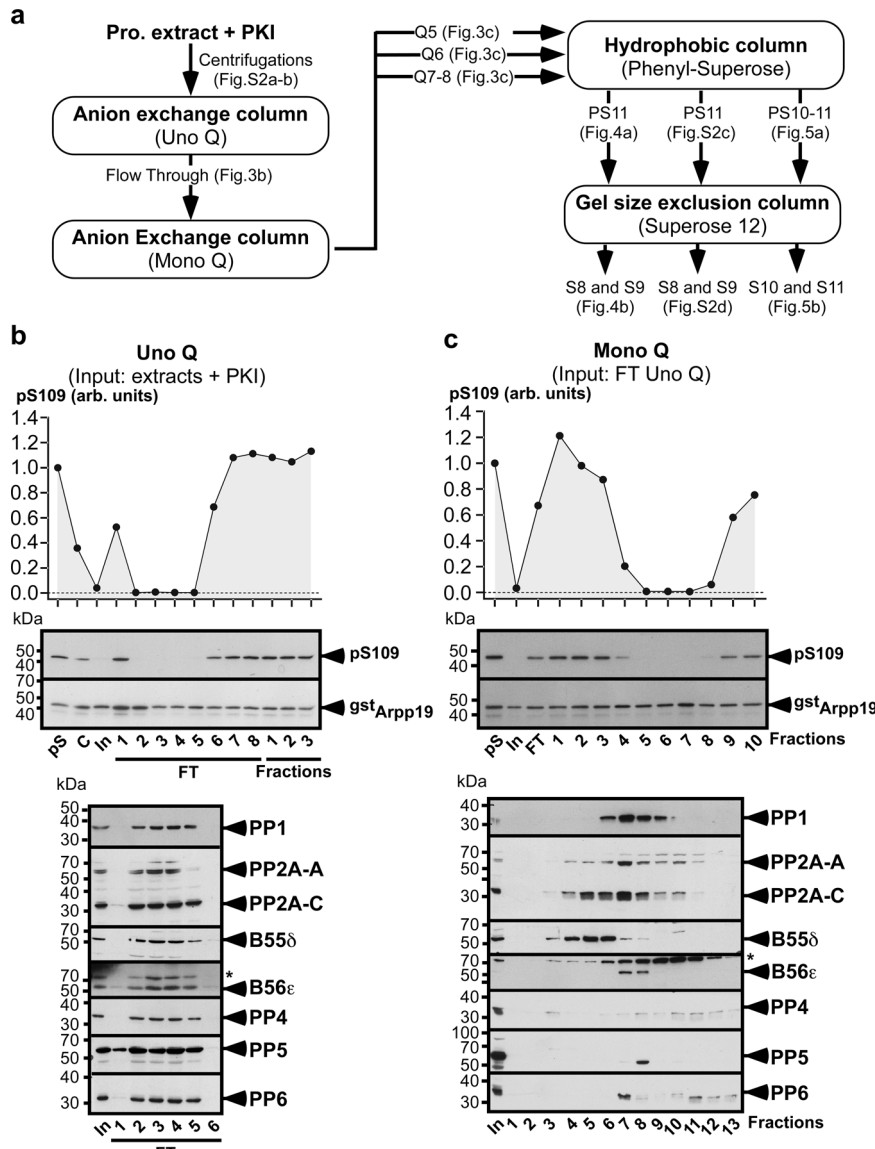

**Fig. 3 Biochemical isolation of S109-phosphatase from prophase extracts - Analysis of the output fractions of Uno Q and Mono Q columns. a** Protocol of S109-phosphatase biochemical isolation. 20,000 prophase oocytes were lysed, centrifuged and fractionated by 4 successive steps of chromatography: two anion exchange columns (Uno Q and Mono Q), one hydrophobic column (Phenyl-Superose) and one size exclusion column (Superose 12). PKI-supplemented extracts from prophase oocytes were fractionated by Uno Q (**b**) and then Mono Q (**c**). S109-phosphatase activity was assayed in each fraction using pS109-GST-Arpp19 as a substrate (pS: phosphorylated substrate). S109 phosphorylation of GST-Arpp19 (pS109) and total GST-Arpp19 (gstArpp19) were analyzed by western blot using respectively phospho-S109-Arpp19 and GST antibodies. Fractions were western blotted with antibodies against the catalytic subunits of PP1, PP2A (PP2A-C), PP4, PP5 and PP6, PP2A scaffold subunit A (PP2A-A) and PP2A regulatory subunits B55δ and B56ε. *: non-specific protein recognized by the anti-B56ε antibody. "C": control extracts before PKI addition. "In": input sample supplemented with PKI and loaded on the column. "FT": flow-through. arb. units: arbitrary units. **b** Uno Q. FT and elution profile (fractions 1–3) of S109-phosphatase activity after Uno Q column and western blot analysis of fractions 1 to 6 of FT. (**c**) Mono Q. Fractions 2 to 5 of the Uno Q column FT (see **b**) were pooled and loaded on the column. Elution profile of S109-phosphatase activity after Mono Q column and western blot analysis of fractions 1 to 13. kDa: kiloDalton. Source data are provided as a Source Data file.

PP2A-A subunit, were recovered in the 40% precipitate (Fig. 2f). In contrast, PP1 and PP5 were mostly detected in the 60% precipitate (Fig. 2f). Hence, the ammonium sulfate fractionation supports our previous results showing that S109-phosphatase is counterbalanced by PKA and indicates that it corresponds neither to PP1 nor to PP5.

**Biochemical identification of S109-phosphatase as PP2A-B55α/δ.** To identify the S109-phosphatase, a chromatography-based procedure was undertaken. PKI-supplemented prophase extracts

were ultracentrifuged (Supplementary Fig. 2a and b) and the supernatant was fractionated by four successive steps of chromatography (Fig. 3a): two anion exchange columns, Uno Q and Mono Q, a hydrophobic column, Phenyl-Superose, and a size-exclusion column, Superose 12. At each step, S109-phosphatase activity was assayed in the fractions using pS109-GST-Arpp19 as a substrate. Fractions were western blotted with various antibodies against phosphatases and some were subject to LC-MS/MS experiments.

Following the Uno Q column, S109-phosphatase activity was recovered in the flow-through (Fig. 3b), as well as PP2A-B55δ, PP2A-B56ε and the catalytic subunits of PP1, PP4, PP5 and PP6

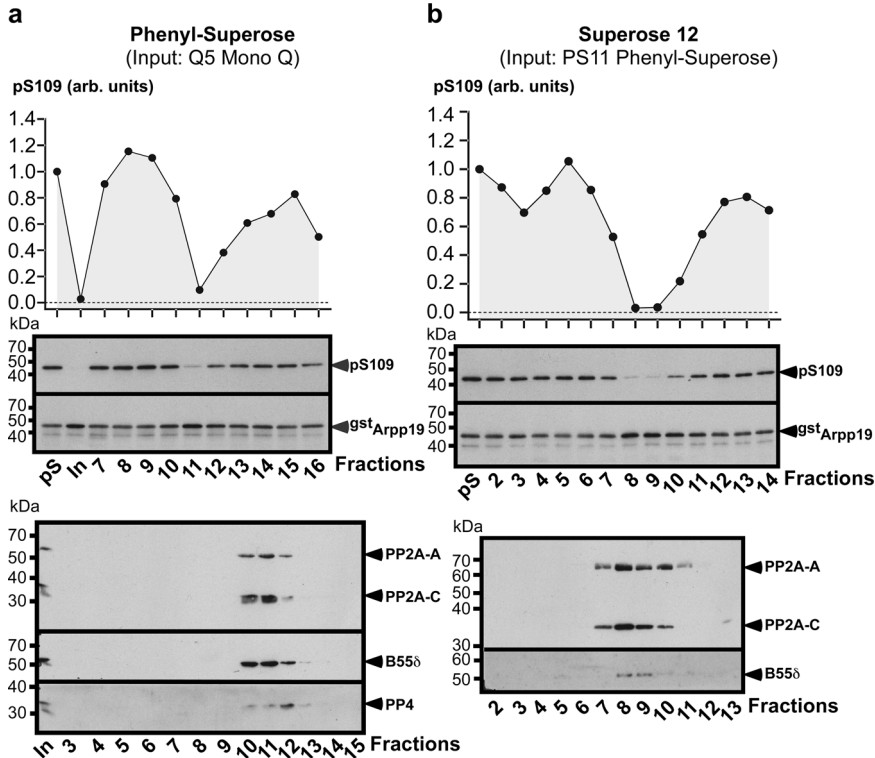

**Fig. 4 Biochemical isolation of S109-phosphatase from prophase extracts—Separation of fraction 5 from the Mono Q column with Phenyl-Superose and Superose 12 columns.** Continuation of experiment illustrated in Fig. 3. "pS": phosphorylated pS109-GST-Arpp19 substrate. "In": input sample loaded on the column. kDa: kiloDalton. arb. units: arbitrary units. **a** Phenyl-Superose. Fraction 5 from the Mono Q column (see Fig. 3c) was loaded on the column. Elution profile of S109-phosphatase activity after Phenyl-Superose column and western blot analysis of fractions 3–15 with antibodies directed against catalytic subunits of PP2A (PP2A-C) and PP4, PP2A scaffold subunit A (PP2A-A) and PP2A regulatory subunit B55δ. **b** Superose 12. Fraction 11 from the Phenyl-Superose column (see **a**) was loaded on the column. Elution profile of S109-phosphatase activity after Superose 12 column and western blot analysis of fractions 2–13 with antibodies directed against PP2A scaffold subunit (PP2A-A), PP2A catalytic subunit (PP2A-C) and PP2A regulatory subunit B55δ. Source data are provided as a Source Data file.

(Fig. 3b). Fractions 2 to 5 in the flow-through were pooled and loaded on a Mono Q column. S109-phosphatase activity was predominantly eluted in fractions 5–8 (Fig. 3c). Fractions 5 and 6 contain PP2A-B55δ, whereas PP4, PP5 and PP6 were not detectable in these fractions (Fig. 3c, Supplementary Table 1). Some PP1 was present in fraction 6 but not in fraction 5 (Fig. 3c, Supplementary Table 1). S109-phosphatase activity was also detected in fractions 7 and 8 that contain various amounts of PP1, PP2A-B55δ, PP2A-B56ε, PP4, PP5, PP6, and PP2C (Fig. 3c and Supplementary Table 1). Hence, PP2A-B55δ is predominant in fractions 5 and 6 whereas fractions 7 and 8 are enriched in PP1, PP2A-B56ε, PP4, PP5, PP6 and PP2C, with few PP2A-B55δ. Fraction 6 corresponds to an intermediary between both groups. The procedure to isolate S109-phosphatase was pursued by separately analyzing fraction 5, fraction 6 and a pool of fractions 7 and 8 (Fig. 3a).

Fractions 5 and 6 were subjected to Phenyl-Superose chromatography. For both fractions, S109-phosphatase activity was eluted in a single fraction, fraction 11, where the main phosphatase present is PP2A-B55α/δ with traces of PP4 (Fig. 4a, Supplementary Fig. 2c and Supplementary Table 1). Consistently with the absence of PP1, PP5 and PP6 in the input, these three phosphatases were undetectable in fraction 11 by LC-MS/MS analysis. PP1, initially present in fraction 6 of the Mono Q, was not recovered in any fraction from 7 to 16 of the Phenyl-Superose column (Supplementary Table 1). Therefore, PP2A-B55α/δ is the major phosphatase present in the fraction 11 that contains S109-phosphatase activity. This fraction was then loaded on a Superose 12 column. The maximal activity of S109-phosphatase was recovered in fractions 8

and 9 where the only phosphatase present was PP2A-B55α/δ (Fig. 4b, Supplementary Fig. 2d and Supplementary Table 1).

Fractions 7 and 8 from the Mono Q column were pooled and subject to Phenyl-Superose chromatography. S109-phosphatase activity was eluted in fractions 10–12 (Fig. 5a). PP2A-B55δ, PP2A-B56α/β/ε and PP2C were recovered in these fractions while PP1, PP4, PP5, and PP6 were not detected or were present in trace amounts (Fig. 5a and Supplementary Table 1). Fractions 10 and 11 were pooled and loaded on a gel filtration Superose 12 column. S109-phosphatase activity was recovered in fractions 7–13, with a peak in fractions 10 and 11 (Fig. 5b). The phosphatases contained in these fractions were PP2A-B55δ, PP2A-B56α/β/ε and PP2C (Fig. 5b and Supplementary Table 1).

Hence, the biochemical procedure to isolate S109-phosphatase activity led to PP2A-A/C and excluded PP1, PP4, PP5 and PP6. As PP2C is insensitive to OA[33], this phosphatase is unlikely responsible for S109-phosphatase activity. Remarkably, S109-phosphatase activity was carried by PP2A-A/C associated with either B55α/δ or B56α/β/ε. The biochemical isolation procedure was performed 4 times, with 3 of these 4 experiments analyzed by LC-MS/MS. Experiment 1 was presented in Figs. 3 to 5, Supplementary Fig. 2c and 2d and Supplementary Table 1, while Supplementary Tables 2 and 3 summarize the two other experiments. In experiment 2, fractions 5 to 8 from the Mono Q column were pooled before Phenyl-Superose. In experiment 3, S109-phosphatase was eluted in a single fraction from the Mono Q column. The final fractions from the Superose 12 column containing S109-phosphatase activity were analyzed by LC-MS/MS analysis (Supplementary Tables 2 and 3). PP2A-B56 isoform

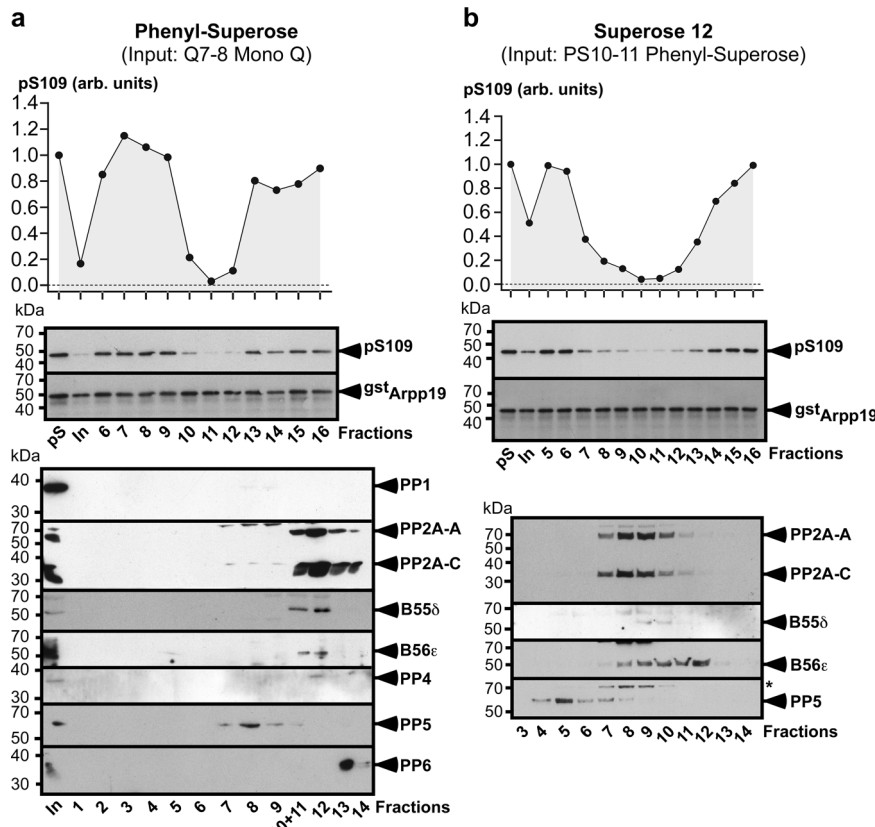

**Fig. 5 Biochemical isolation of S109-phosphatase from prophase oocyte extracts - Separation of fractions 7 and 8 from the Mono Q column with Phenyl-Superose and Superose 12 columns.** Continuation of experiment illustrated in Fig. 3. "pS": phosphorylated pS109-GST-Arpp19 substrate. "In": input sample loaded on the column. kDa: kiloDalton. arb. units: arbitrary units. **a** Phenyl-Superose. Fractions 7 and 8 from the Mono Q column (see Fig. 3c) were pooled and loaded on the column. Elution profile of S109-phosphatase activity after Phenyl-Superose column and western blot analysis of fractions 1–14 with antibodies directed against the catalytic subunits of PP1, PP2A (PP2A-C), PP4, PP5, and PP6, PP2A scaffold subunit (PP2A-A) and PP2A regulatory subunits B55δ and B56ε. **b** Superose 12. Fraction 10 and 11 from the Phenyl-Superose column (see **a**) were pooled and loaded on the column. Elution profile of S109-phosphatase activity after Superose 12 column and western blot analysis of fractions 3–14 with antibodies directed against the catalytic subunits of PP2A (PP2A-C) and PP5, PP2A scaffold subunit (PP2A-A) and PP2A regulatory subunits B55δ and B56ε. *non-specific protein recognized by the anti-PP5 antibody. Source data are provided as a Source Data file.

was recovered in the final fractions of experiment 2, in which fractions 5–8 from the Mono Q column were pooled, whereas it was barely detectable in experiment 3 where S109-phosphatase was found in a single fraction after the Mono Q column. This argues for B56 contaminating S109-phosphatase fractions when the Mono Q fractions are pooled. In contrast, PP2A-B55 isoform was recovered in all experiments (Supplementary Tables 2 and 3), arguing for PP2A-B55α/δ being the main phosphatase carrying S109-phosphatase activity.

**Arpp19 is dephosphorylated at S109 by purified PP2A-B55δ but not by purified PP1.** To support the results of the biochemical procedure, we assayed Arpp19 dephosphorylation with purified enzymes. *Xenopus* catalytic His-PP1α and His-B55δ were expressed in prophase oocytes injected with their respective encoding mRNAs and affinity-purified using Co-beads (Fig. 6a). Both purified PP1 and PP2A-B55δ enzymes were used in in vitro dephosphorylation assays. PP2A-B55δ is known to dephosphorylate Arpp19 at S67[21]. We assessed that purified PP2A-B55δ was able to dephosphorylate Arpp19 at S67. A mutant form of Arpp19 that cannot be phosphorylated at S109, S109A-GST-Arpp19, was phosphorylated at S67 and used as a substrate. As expected, PP2A-B55δ efficiently dephosphorylated Arpp19 at S67 in contrast to PP1 (Fig. 6b and c). We then used pS109-GST-Arpp19 as a substrate. The reaction kinetics

revealed that PP2A-B55δ efficiently dephosphorylates Arpp19 at S109 in contrast to PP1 whose rate of S109 dephosphorylation is very slow (Fig. 6d and e). Therefore, Arpp19 dephosphorylation observed in extracts or chromatography fractions can be recapitulated with purified enzymes.

**PP2A-B55δ dephosphorylates Arpp19 at S109 in prophase extracts.** If PP2A-B55δ corresponds to the S109-phosphatase, it should be active in prophase-arrested oocytes. Since PP2A-B55δ is known to dephosphorylate Arpp19 at S67[21], its activity was assayed in prophase extracts with Arpp19 phosphorylated at both S67 and S109 (pS67-pS109-GST-Arpp19). This substrate allows monitoring the concomitant dephosphorylation of S109 and S67, respectively catalyzed by S109-phosphatase and PP2A-B55δ. As expected, S109-phosphatase dephosphorylated pS67-pS109-GST-Arpp19 at S109 (Fig. 7a and b) in a PKI- and OA-dependent manner (Supplementary Fig. 3a). Interestingly, pS67-pS109-GST-Arpp19 was also dephosphorylated at S67, strongly arguing for PP2A-B55δ being active in prophase extracts (Fig. 7c and d). This event was prevented by OA but was insensitive to PKI (Supplementary Fig. 3b), an expected result as S67 is targeted by Gwl but not by PKA[24]. Since S67-phosphorylated Arpp19 inhibits PP2A-B55δ activity, it should impair the dephosphorylation of S109[13,14]. Indeed, the kinetic parameters show that Arpp19 is not dephosphorylated at S109 as long as S67 remains phosphorylated

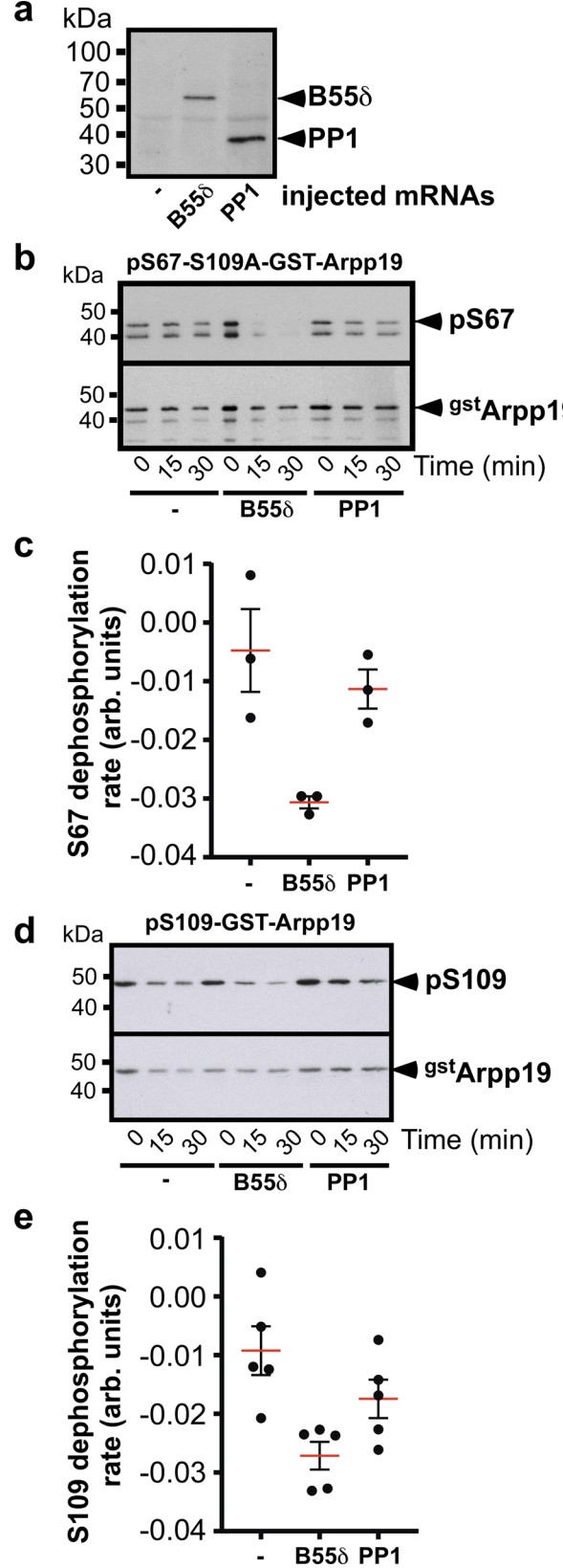

**Fig. 6 In contrast to PP1, purified PP2A-B55δ dephosphorylates Arpp19 at both S109 and S67. a** Co-beads were incubated with extracts of prophase oocytes injected or not with mRNAs coding either His-PP1 or His-B55δ and then analyzed by western blot using an anti-Histidine antibody. The experiment was repeated 5 times with similar results. **b**–**c** S67-phosphatase activity of either PP1 or PP2A-B55δ coupled to Co-beads using pS67-S109A-GST-Arpp19 as a substrate. S67 phosphorylation of S109A-GST-Arpp19 (pS67) and total GST-S109A-Arpp19 ($^{gst}$Arpp19) were western blotted using respectively phospho-S67-Arpp19 and GST antibodies. Time course of one representative experiment (**b**) and quantified rate of S67 dephosphorylation from 3 independent experiments (**c**). **d**–**e** S109-phosphatase activity of either PP1 or PP2A-B55δ coupled to Co-beads using pS109-GST-Arpp19 as a substrate. S109 phosphorylation of GST-Arpp19 (pS109) and total GST-Arpp19 ($^{gst}$Arpp19) were western blotted using respectively phospho-S109-Arpp19 and GST antibodies. Time course of one representative experiment (**d**) and quantified rate of S109 dephosphorylation from 5 independent experiments (**c**). For quantifications, data are shown as mean (red bars) ± SEM. Each dot represents one experiment. kDa: kiloDalton. arb. units: arbitrary units. Source data are provided as a Source Data file.

To ascertain that PP2A-B55δ is responsible for Arpp19 dephosphorylation at S109, this phosphatase was specifically depleted from prophase extracts using its specific inhibitor, S67-phosphorylated Arpp19, coupled to GSH-beads. To avoid S67 dephosphorylation, GST-Arpp19 was thiophosphorylated with ATP-γS (tpS67-GST-Arpp19)[24]. As a positive control, microcystin-beads known to inhibit all PP2A-A/C isoforms by direct binding were used[34]. After incubation with either tpS67-GST-Arpp19-beads or microcystin-beads, supernatants and beads were separated. As expected from the pan-specificity of microcystin, PP2A subunits A, C, B56ε, and B55δ were recovered on the microcystin-beads (Supplementary Fig. 3c). In contrast, given the specificity of tpS67-GST-Arpp19, B55δ was strongly enriched in tpS67-GST-Arpp19-beads, whereas B56ε was barely detectable (Supplementary Fig. 3c). S109-phosphatase activity was then assayed in the supernatants. In the absence of PKI, Arpp19 phosphorylation at S109 was increased in PP2A-depleted supernatants with microcystin-beads (Fig. 7e and f), showing that PKA actively phosphorylates Arpp19 at S109 in the absence of PP2A. Inhibiting PKA with PKI led to a strong S109 dephosphorylation of Arpp19 in prophase extracts (Fig. 7e and f). In contrast, such a dephosphorylation was no longer observed in the microcystin-depleted supernatants (Fig. 7e and f). Importantly, the specific depletion of PP2A-B55δ with tpS67-GST-Arpp19-beads similarly abolished Arpp19 dephosphorylation at S109 (Fig. 7g and h). Altogether, these results indicate that PP2A-B55δ behaves as the S109-phosphatase in prophase extracts: it is active in dephosphorylating Arpp19 and counterbalanced by PKA.

**PP2A-B55δ dephosphorylates Arpp19 at S109 in intact oocytes.** We first ascertained whether Arpp19 phosphorylation at S109 is also subject to a turnover in intact oocytes. We used as S109-phosphatase substrate a GST-truncated form of Arpp19, corresponding to its C-terminal part (68-117) (Cter-GST-Arpp19). This peptide contains S109 but lacks S67 that is phosphorylated by Gwl during meiosis resumption[24]. In vitro, Cter-GST-Arpp19 was efficiently phosphorylated at S109 by PKA and efficiently dephosphorylated at S109 when incubated in PKI-supplemented prophase extracts (Supplementary Fig. 4a and 4b). Moreover, unlike full-length Arpp19 protein[3,24], Cter-GST-Arpp19 did not interfere with meiotic maturation (Supplementary Fig. 4c and 4d). Prophase oocytes were injected with Cter-GST-Arpp19 and either stimulated with progesterone or injected with PKI or OA. Cter-GST-Arpp19

(Fig. 7a–d). Since Greatwall is not active in prophase extracts, Arpp19 cannot be rephosphorylated at S67, accounting for the loss of its inhibitory effect toward PP2A-B55δ that can eventually dephosphorylate S109. Altogether, our results indicate that PP2A-B55δ is active in prophase oocytes and not affected by PKA.

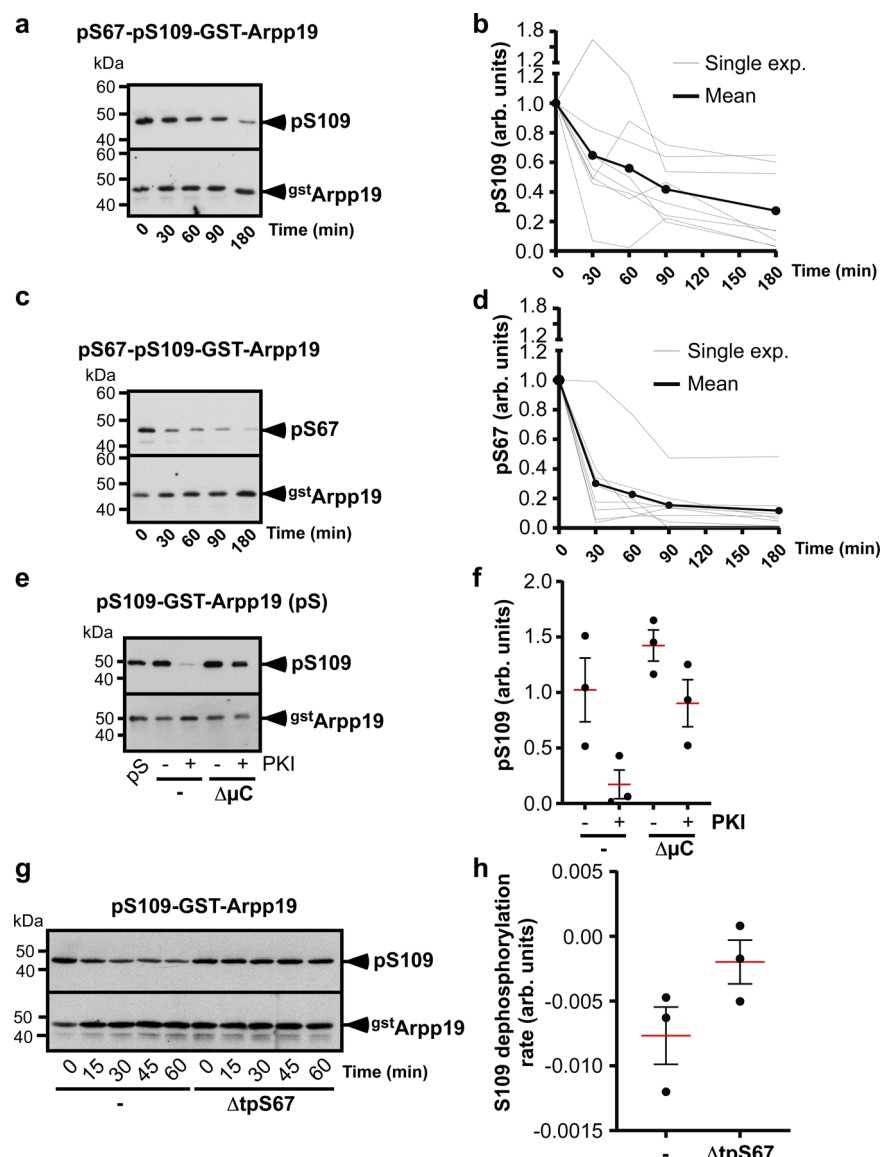

**Fig. 7 Depletion of PP2A-B55δ impairs Arpp19 dephosphorylation at S109 in prophase extracts.** Prophase extracts were incubated with GST-Arpp19 phosphorylated at both S67 and S109 (pS67-pS109-GST-Arpp19) in the absence of ATP. S109-phosphatase (**a–b**) and S67-phosphatase activities (**c–d**) were assayed by western blot using phospho-S109-Arpp19, phospho-S67-Arpp19 and GST antibodies. **a, c** Representative time course experiments. **b, d** Quantifications of 8 time-course independent experiments. Each experiment is represented by a gray curve (Single exp.), and the mean of the 8 replicates by the black curve. arb. units: arbitrary units. **e–f** Prophase extracts were incubated or not with microcystin-beads (μC). After beads removal, control (−) or microcystin-depleted prophase extracts (ΔμC) were supplemented or not with PKI and assayed for S109-phosphatase activity using pS109-GST-Arpp19 as a substrate (pS: phosphorylated substrate). S109 phosphorylation of GST-Arpp19 (pS109) and total GST-Arpp19 (gstArpp19) were western blotted using respectively phospho-S109-Arpp19 and GST antibodies. One representative experiment (**e**) and quantifications of S109 phosphorylation from 3 independent experiments (**f**). **g–h** Prophase extracts were incubated with GSH-beads coupled or not with GST-Arpp19 thiophosphorylated at S67 (tpS67) in the absence of ATP. After beads removal, control (−) or tpS67-depleted prophase extracts (ΔtpS67) were assayed for S109-phosphatase activity using pS109-GST-Arpp19 as a substrate. S109 phosphorylation of GST-Arpp19 (pS109) and total GST-Arpp19 (gstArpp19) were western blotted using respectively phospho-S109-Arpp19 and GST antibodies. One representative time-course experiment (**g**) and quantified rate of S109 dephosphorylation from 3 independent experiments (**h**). For quantifications in (**f**) and (**h**), data are shown as mean (red bars) ± SEM. Each dot represents one experiment. kDa: kiloDalton. arb. units: arbitrary units. Source data are provided as a Source Data file.

was then isolated by pull-down at various times before Cdk1 activation. As shown in Fig. 8a and b, Cter-GST-Arpp19 was in vivo phosphorylated at S109 in prophase oocytes and was partly dephosphorylated in response to progesterone, thus reproducing the dephosphorylation of endogenous Arpp19 observed upon hormonal stimulation[3]. In the absence of progesterone, PKI injection induced the full dephosphorylation of Cter-GST-Arpp19 at S109 while OA injection increased the S109 phosphorylation level (Fig. 8a and b). Hence, S109-phosphatase is an OA-sensitive phosphatase that is

active in prophase oocytes, the phosphorylation level of Arpp19 at S109 resulting from a balance between PKA and its opposing phosphatase, in favor of PKA.

Whether PP2A-B55δ also dephosphorylates Arpp19 at S109 in intact oocytes was ascertained using its specific inhibitor, tpS67-GST-Arpp19. To avoid any side-specific effect due to S109 phosphorylation, a non-phosphorylatable Arpp19 mutant, in which S109 was replaced by alanine, was used (S109A-tpS67-GST-Arpp19)[26]. S109A-tpS67-GST-Arpp19 was injected in

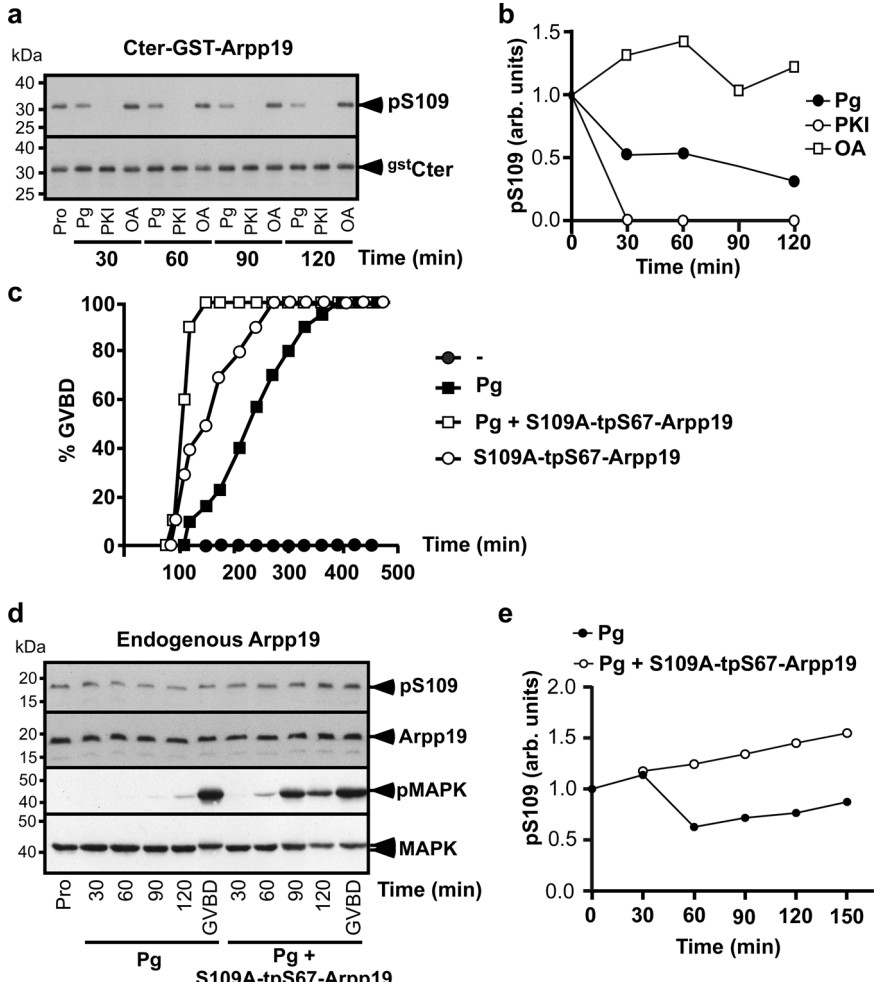

**Fig. 8 Inhibition of PP2A-B55δ prevents Arpp19 dephosphorylation at S109 in intact oocytes. a** Prophase oocytes (Pro) were injected with Cter-GST-Arpp19 (gstCter) and then stimulated with progesterone (Pg) or injected with either PKI or okadaic acid (OA) (time zero). Oocytes were collected at the indicated times and Cter-GST-Arpp19 was isolated by pull-down. S109 phosphorylation of Cter-GST-Arpp19 (pS109) and total Cter-GST-Arpp19 (gstCter) were western blotted using respectively phospho-S109-Arpp19 and GST antibodies. **b** S109 phosphorylation of the experiment illustrated in (**a**) was quantified. **c–e** Prophase oocytes were injected or not with S109A-GST-Arpp19 thiophosphorylated at S67 (S109A-tpS67-Arpp19) and then stimulated or not with progesterone (Pg). **c** GVBD time-course. **d** Same experiment as (**c**). Oocytes were collected in prophase (Pro) or at the indicated times after Pg addition and western blotted for S109-phosphorylated endogenous Arpp19 (pS109), total endogenous Arpp19, phosphorylated MAPK (pMAPK) and total MAPK. (**e**) Quantification of S109 phosphorylation of endogenous Arpp19 from the experiment illustrated in (**d**). kDa: kiloDalton. arb. units: arbitrary units. Source data are provided as a Source Data file.

prophase oocytes. As expected[8,24,35], this protein promoted meiosis resumption independently of progesterone (Fig. 8c). Cdk1 activation was ascertained by western blotting phosphorylated MAPK (Fig. 8d), a marker of Cdk1 activation[36]. Arpp19 dephosphorylation at S109 was analyzed by western blot (Fig. 8d and e). As previously shown[3], the protein was partially dephosphorylated at S109 within 1 h upon progesterone addition and was then rephosphorylated at this residue before GVBD in control oocytes (Fig. 8d and e). In contrast, Arpp19 was not dephosphorylated at S109 when PP2A-B55δ was specifically inhibited by S109A-tpS67-GST-Arpp19 (Fig. 8d and e). Hence, inhibiting PP2A-B55δ in oocytes is sufficient to abolish Arpp19 dephosphorylation at S109 induced by progesterone. PP2A-B55δ is therefore responsible for Arpp19 dephosphorylation during oocyte meiosis resumption.

**Arpp19 dephosphorylation at S109 by PP2A-B55δ is independent of Cdk1.** Cdk1 activation depends on PP2A-B55δ inhibition thanks to Arpp19 phosphorylation at S67 that occurs at

GVBD[24]. This negative action of PP2A-B55δ is controlled by Cdk1 itself, which indirectly activates Gwl, constituting a positive feedback loop[5]. Remarkably, our present results reveal that PP2A-B55δ is the phosphatase that dephosphorylates Arpp19 at S109 upstream Cdk1 activation, as early as 1 h after progesterone stimulation (Fig. 8d and e), a necessary event to promote meiotic resumption[3]. Hence, both steps, critical for Cdk1 activation, involve PP2A-B55δ and Arpp19 but occur sequentially over time. To ensure that the early action of PP2A-B55δ is disconnected from its late function, we investigated whether Arpp19 dephosphorylation at S109 is independent of Cdk1 activity. Prophase oocytes were injected with a specific inhibitor of Cdk1[37], the protein p21Cip1 and were then stimulated with progesterone. In control oocytes, Arpp19 was phosphorylated at S109 in prophase, partially dephosphorylated at S109 within 1 h after progesterone addition and then rephosphorylated at this residue just before GVBD (Fig. 9a). P21Cip1 injection efficiently prevented Cdk1 activation in response to progesterone (Fig. 9a). Under these conditions, Arpp19 was still dephosphorylated at S109 in response to progesterone and was then maintained in its partially

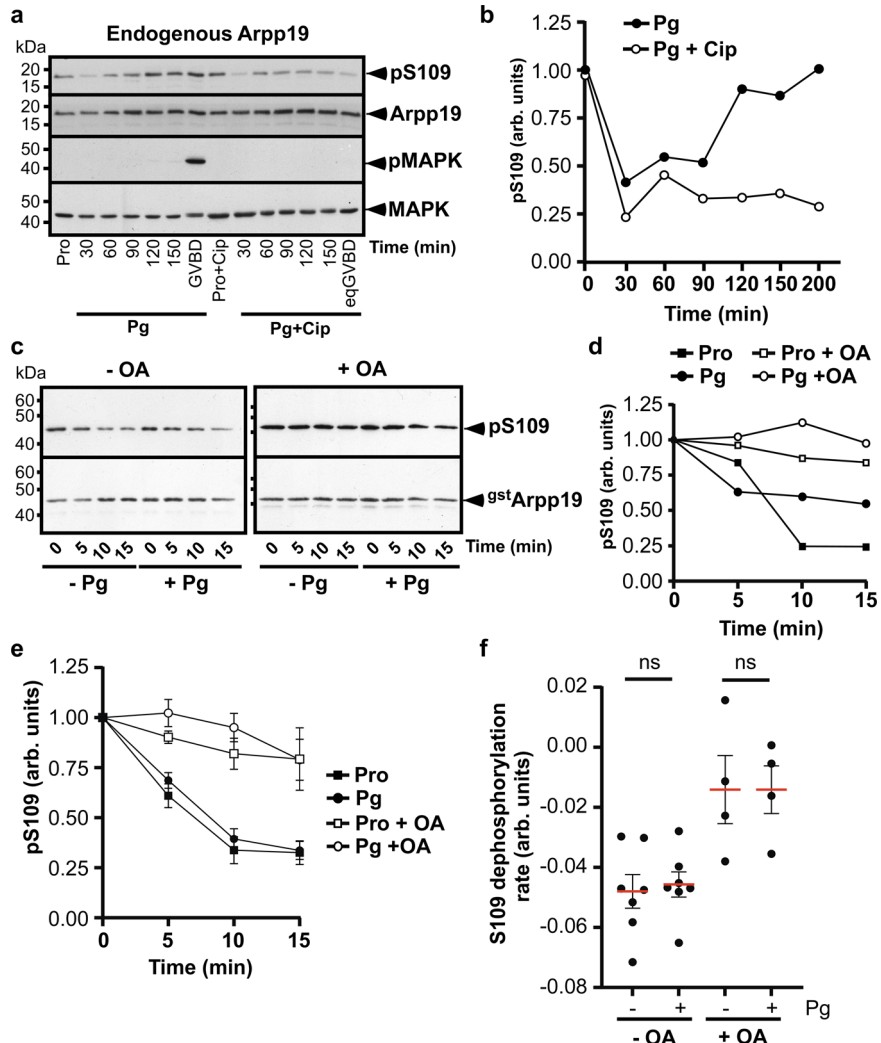

**Fig. 9 Level of PP2A-B55δ activity in intact oocytes. a** Prophase oocytes (Pro) were injected or not with p21Cip1 (Cip) and then stimulated with progesterone (Pg). GVBD occurred at 200 min in control oocytes and was prevented by Cip injection. Oocytes were collected at the indicated times after Pg addition and western blotted with antibodies against S109-phosphorylated Arpp19 (pS109), total Arpp19, phosphorylated MAPK (pMAPK) and total MAPK. **b** S109 phosphorylation of the experiment illustrated in (**a**) was quantified. **c** Extracts from either prophase oocytes (−Pg) or oocytes stimulated for 1 h with progesterone (+Pg) were supplemented or not with 10 μM okadaic acid (OA) in the absence of ATP. The activity of S109-phosphatase was assayed at the indicated times, using pS109-GST-Arpp19 as a substrate. S109 phosphorylation of GST-Arpp19 (pS109) and total GST-Arpp19 (gstArpp19) were western blotted using respectively phospho-S109-Arpp19 and GST antibodies. **d** Quantification of S109 phosphorylation of GST-Arpp19 from the representative experiment illustrated in (**c**). **e** Quantification of S109 phosphorylation of GST-Arpp19 from 7 independent time-course experiments. S109-phosphatase activity was assayed and quantified as in (**c–d**). Four experiments were performed with or without OA and 3 experiments without OA. For each condition, data are shown as mean (red bars) ± SEM. **f** Quantified rates of S109 dephosphorylation from the 7 independent experiments, described in (**e**). Four experiments were performed with or without OA and 3 experiments without OA. Data are shown as mean (red bars) ± SEM. Each dot represents one experiment. $P$ values were obtained by a two-tailed paired Student $t$ test. $P > 0.05$: non-significant (ns). kDa: kiloDalton. arb. units: arbitrary units. Source data are provided as a Source Data file.

dephosphorylated state (Fig. 9a and b). This result indicates that PP2A-B55δ actively dephosphorylates Arpp19 at S109 well before and independently of Cdk1 activation. Both actions of PP2A-B55δ are therefore temporally and functionally disconnected in the oocyte.

**PP2A-B55δ activity is not regulated in the early stages of meiosis resumption.** In prophase oocytes, Arpp19 phosphorylation at S109 depends on a balance between PKA and PP2A-B55δ in favor of PKA. Arpp19 dephosphorylation could be either promoted by the sole inhibition of PKA or by the concomitant up-regulation of PP2A-B55δ activity. To address this

issue, we assayed PP2A-B55δ activity in extracts from oocytes either in prophase or collected 1 h after progesterone stimulation. As expected, endogenous Arpp19 was partially dephosphorylated at S109 at that time (Supplementary Fig. 5). PP2A-B55δ substrate, pS109-GST-Arpp19, was added to oocyte lysates in the presence or in the absence of OA, and its dephosphorylation was ascertained as a function of time. As shown in Fig. 9c and d, Arpp19 was dephosphorylated in an OA-dependent manner, at a similar rate in extracts from oocytes in prophase or stimulated by progesterone. The experiment was performed 7 times using different females (Fig. 9e and f). No statistical difference was observed regarding the S109 dephosphorylation level nor the rates of dephosphorylation by PP2A-B55δ at both stages. Therefore,

PP2A-B55δ activity is not up-regulated by progesterone at the early stages of meiosis resumption and Arpp19 dephosphorylation at S109 solely depends on PKA downregulation.

## Discussion

In all vertebrates, a high level of PKA activity arrests oocytes in prophase of the first meiotic division for long periods. In *Xenopus*, Arpp19 phosphorylated at S109 by PKA plays an important part in maintaining the prophase block[3]. Progesterone releases this block by decreasing cAMP levels and PKA activity within 1 h, resulting in a partial dephosphorylation of Arpp19 at S109 sufficient to trigger a signaling pathway that leads, 3–5 h later, to Cdk1 activation[3,38–40]. We have now uncovered the phosphatase responsible for Arpp19 dephosphorylation at S109, unexpectedly finding it to be the same PP2A isoform that opposes Cdk1 during M-phase entry.

PP1 was initially an attractive candidate to oppose the action of PKA during oocyte prophase arrest. PKA and PP1 antagonistically regulate the activity of many physiological effectors involved in cell cycle regulation. In particular, PP1 inhibition blocks progesterone and PKI-induced meiotic maturation in *Xenopus* oocytes. PP1 was thus proposed to dephosphorylate a PKA substrate, still unknown at that time[41]. PP1 also positively regulates mitosis by contributing to Cdc25C activation and by controlling structural events such as centrosome splitting, spindle assembly and microtubules-kinetochores attachment[42]. PP1 activity is then repressed at mid-mitosis by Cdk1 phosphorylation and reactivated at the end of mitosis where it contributes to the dephosphorylation of Cdk1 substrates[42–44]. Nevertheless, we have ruled out the possibility that PP1 is responsible for Arpp19 dephosphorylation on S109. PP1 did not associate with S109-phosphatase activity in the ammonium sulfate precipitates nor after the Mono Q column and the purified enzyme was not efficient in dephosphorylating Arpp19 at S109 in vitro.

All our in vitro or in vivo approaches demonstrate that the phosphatase that acts on Arpp19 at S109 is a PP2A isoform. We were further able to determine that this activity is predominantly carried by the B55 subunit, i.e., the same isoform that participates as an essential component of the M-phase switch. We detected B55, in some cases accompanied by B56, in the active S109-phosphatase fractions obtained from all three biochemical isolation procedures analyzed by western blot and LC-MS/MS, as well as in an initial isolation experiment monitored by western blot only. While PP2A-B55 is thus clearly the main phosphatase responsible for S109 dephosphorylation of Arpp19, PP2A-B56 can also potentially contribute, at least in extracts. In contrast, inhibiting specifically PP2A-B55δ using S67-phosphorylated Arpp19 in intact oocytes fully abolishes S109 dephosphorylation of endogenous Arpp19 induced by progesterone and even increases its phosphorylation level. These experiments establish that in vivo, PP2A-B55 is the physiological phosphatase that dephosphorylates Arpp19 at S109. The minor contribution of PP2A-B56 in oocyte extracts could result from its release from discrete subcellular compartments during extract preparation. It has been established that PP2A-B56 localizes specifically to kinetochores and centromeres, participating in chromatid-microtubule interactions and silencing of the spindle assembly checkpoint[45,46]. Likely relating to this M-phase role, PP2A-B56 is involved in the metaphase II-arrest of *Xenopus* oocyte by regulating the APC/C inhibitor, XErp1[47,48]. These M-phase roles of PP2A-B56 are not directly connected to Cdk1 regulation and do not involve Arpp19. Accordingly, Arpp19 lacks the two known binding motifs for PP2A-B56 but includes bipartite recognition determinants for PP2A-B55[49–51].

Each of the four B subfamilies comprises several isoforms with very closely related sequences, no discernible differences in their substrate binding pockets and substantial substrate specificity overlap[52]. In the present study, our biochemical isolation procedures identify B55δ as being associated with PP2A-A/C that dephosphorylates Arpp19 at S109. B55α was also detected by LC-MS/MS experiment. Since we have no specific anti-B55α antibody, we cannot exclude that S109-phosphatase activity is a mixture of PP2A-B55α and PP2A-B55δ. Nevertheless, in *Xenopus* oocytes, B55δ accounts for 70% or more of the total B55 subunits, with B55α and B55β antibodies failing to detect endogenous proteins, due to their low expression level[8,53]. Moreover, Arpp19 is not detected within the protein interactome of PP2A-B55α[54]. We therefore ascribe the majority of the phosphatase responsible for dephosphorylating Arpp19 at S109 to the PP2A subset containing the B55δ subunit. Similarly, in *Xenopus* egg extracts, PP2A-B55δ is the key phosphatase isoform that acts on Cdk1 substrates, a property not shared by the other *Xenopus* PP2A-B55 holoenzymes[8,53] and is inhibited by Arpp19 phosphorylated at S67[13].

Our results revealed that Arpp19 phosphorylation at S109 is subject to dynamic turnover in prophase-arrested oocytes. The two mutually antagonistic enzymes, PKA and PP2A-B55δ, work simultaneously, with the action of PP2A-B55δ being swamped by PKA. This contrasts with other systems in which PKA plays a critical role in phosphate removal by regulating protein phosphatase activities[55], for example by phosphorylating B56 and activating PP2A-B56δ in human brain[56]. The simultaneous activities of PKA and PP2A-B55δ result in a futile phosphorylation cycle. This not only renders impossible a full switch-like interconversion of Arpp19 phosphorylation state, but also allows the two opposed active enzymes to carry important functions independently of each other. As such, PKA certainly targets substrates other than Arpp19, important to keep oocytes arrested in prophase, such as Cdc25[57] or, similarly to Wee1b in mouse[58], Myt1. Meantime, the sustained activity of PP2A-B55δ also ensures the stability of the prophase arrest by impeding Cdk1 activation and its substrates phosphorylation.

Remarkably, both Arpp19 sites important for the control of meiosis resumption, S109 and S67, are dephosphorylated by a unique phosphatase, PP2A-B55δ. The function of Arpp19 phosphorylation at S67 in converting this protein into a PP2A-B55δ inhibitor is well documented, whereas the action of S109 phosphorylation of Arpp19 in ensuring the prophase arrest remains unknown. It is tempting to speculate that, by mirroring the function of S67-phosphorylated Arpp19, S109-phosphorylated Arpp19 regulates one of its regulatory kinases or phosphatases. Such a dual function has been described for two Arpp19-related proteins, DARPP-32 and Arpp16. Depending on its phosphorylation by PKA or by Cdk5 at two distinct sites, DARPP-32 acts as an inhibitor of either PP1 or PKA[59,60]. Arpp16, exclusively expressed in the brain, and Arpp19, ubiquitously expressed, are alternatively spliced variants of the same gene[61,62]. Like Arpp19, Arpp16 is phosphorylated by Gwl at S46 and by PKA at S88[61,63]. As reported for Arpp19, Arpp16 is converted into a strong inhibitor of specific PP2A isoforms when phosphorylated by Gwl[63]. Interestingly, the PKA phosphorylated form of Arpp16 at S88 makes PP2A non-inhibitable[64] and, in contrast to Arpp19, phosphorylation of Arpp16 at S88 and S46 is mutually antagonistic[26,64]. These findings raise the question of whether Arpp19 is a dual-function protein, like DARPP-32 or Arpp16. Although our preliminary attempts failed to show that S109-phosphorylated Arpp19 interacts with PP2A and regulates its activity, this hypothesis deserves deeper investigation. Another possibility is that Arpp19 behaves as a PP1 inhibitor when phosphorylated by PKA, similarly to DARPP-32. In response to progesterone, S109 dephosphorylation of Arpp19 by PP2A-B55δ would activate PP1, an event that positively regulates entry into cell division[41,65]. Although the catalytic subunit of PP1 is

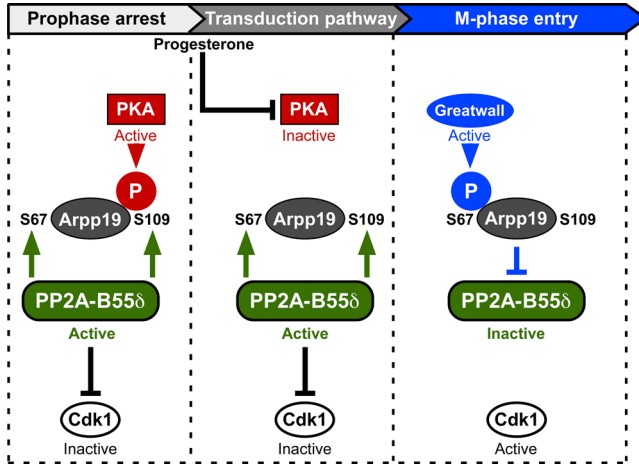

**Fig. 10 A reciprocal regulation of PP2A-B55δ and Arpp19 orchestrates the timing of the first meiotic division.** Prophase arrest (left box): both PP2A-B55δ and PKA are active, resulting in Arpp19 phosphorylation at S109. S109-phosphorylated Arpp19 locks the oocyte in prophase by an unknown mechanism, PKA contributes to this arrest through Arpp19 phosphorylation and possibly other substrates, PP2A-B55δ prevents Cdk1 activation. Transduction pathway (middle box): in response to progesterone, PKA is inhibited while PP2A-B55δ stays active, allowing Arpp19 dephosphorylation. Dephosphorylated Arpp19, possibly together with other dephosphorylated PKA substrates, launches a several hours long transduction pathway. Active PP2A-B55δ prevents Cdk1 activation, generating the time window necessary to set up the cascade of molecular events required for Cdk1 activation. M-phase entry (right box): Gwl is activated by a Cdk1 activity threshold and phosphorylates Arpp19 at S67 Hence, PP2A-B55δ is inhibited and Cdk1 fully activated. This hysteretic switch triggers M-phase entry.

not regulated in vitro by S109-phosphorylated Arpp19[13], this does not exclude the possibility that Arpp19 could control in vivo PP1 through its regulatory subunits and with respect to specific physiological substrates.

Our results highlight how the reciprocal regulation of PP2A-B55δ and Arpp19 orchestrates the timing of the first meiotic division in *Xenopus* oocytes in a 3-step process (Fig. 10). First, S109-phosphorylated Arpp19 and PP2A-B55δ secure the prophase arrest: PP2A-B55δ prevents any protein phosphorylation that could lead to unscheduled Cdk1 activation, while the action of S109-phosphorylated Arpp19 is not yet known. Second, progesterone triggers PKA inhibition while PP2A-B55δ stays active. As a consequence, Arpp19 is dephosphorylated at S109 and launches a several hours signaling pathway that ends with Cdk1 activation. Keeping PP2A-B55δ active prevents premature activation of Cdk1 and thus allows sufficient time for the signaling pathway, especially the translation of new proteins required for meiotic divisions. The third step is a switch conversion, relying on a threshold of a few active Cdk1 molecules. These activate Gwl, which in turn phosphorylates Arpp19 at S67 and inhibits PP2A-B55δ, setting up the bistable switches that govern irreversible M-phase entry[11,28]. Since PP2A-B55δ is inhibited, Arpp19 can be rephosphorylated at S109. Indeed, this is achieved by a kinase distinct from PKA and depending on Cdk1 activity[3]. The role of this new S109 phosphate, if any, is unknown. Hence, we discovered that a double function of PP2A-B55δ and its substrate, Arpp19, governs oocyte meiosis resumption, opening new avenues in the control of oocyte meiosis, but more widely in the control of cell cycle progression and its dysfunction in human pathologies related to fertility and cancer.

## Methods

**Materials.** *Xenopus laevis* adult females (Centre de Ressources Biologiques Xenopes, CNRS, France) were bred and maintained according to current French guidelines in the conventional IBPS aquatic animal facility, with authorization: Animal Facility Agreement: #A75-05-25. All experiments were subject to ethical review and approved by the French Ministry of Higher Education and Research (reference APAFIS#14127-2018031614373133v2).

All reagents, unless otherwise specified, were from Sigma. Okadaic acid (OA), magnetic GSH-beads and Co-beads were purchased from Enzo Life Sciences, Promega and Clontech Laboratories respectively. Uno Q-25 was purchased from Bio-Rad, Mono Q 4.6/100PE, Phenyl-Superose HR5/5 and Superose 12 HR10/30 were from GE Healthcare.

**Xenopus Oocyte handling.** Full-grown prophase oocytes were obtained from unprimed *Xenopus laevis* females as described[66]. Oocytes were microinjected with OA (2 μM), His-K71M-Gwl mRNA (150 μg), His-catalytic PP1 mRNA (30 ng), His-B55δ mRNA (30 ng) or the following recombinant proteins: Cter-GST-Arpp19 (125 ng, 250 ng, or 500 ng), tpS67-S109A-Arpp19 (250 ng), p21Cip1 (35 ng) or PKI (75 ng) in 50 nl final volume per oocyte. A 2 μM progesterone was added to the external medium. Each in vivo experimental condition was applied to groups of 15–30 oocytes. Oocytes were referred to as GVBD when the first pigment rearrangement was detected at the animal pole.

**Western blots.** For all proteins except endogenous Arpp19, an equivalent of 0.6 oocyte was subjected to SDS gel electrophoresis (12, 10 or 8%)[67] and then transferred onto nitrocellulose as described[68]. To visualize endogenous Arpp19 protein, an equivalent of 1.2 oocytes was loaded on 15.5% Tris-Tricine gels[69]. The antibodies directed against the following proteins were used: Arpp19[70] (1:1,000, gift of Dr Angus Nairn, Yale University, USA), S109-phosphorylated Arpp19[3] (1:500), S67-phosphorylated Arpp19[24] (1:1,000), MAP kinase (1:1,000, Santa Cruz SC-154), phosphorylated MAP kinase (1:1,000, Cell Signaling 9106), PKA catalytic subunit (1:1,000, Abcam ab26322), PKI (1:1,000, Abcam ab122816), PP1 catalytic subunit (1:100, gift of Dr M. Bollen, KU Leuven, Belgium), PP2A-C catalytic subunit[71] (1:500), PP2A-A scaffold subunit[71] (1:500), B55δ regulatory subunit[8] (1:500, gift of Dr S. Mochida, Kumamoto University, Japan), B56ε regulatory subunit[8] (1:500, gift of Dr S. Mochida, Kumamoto University, Japan), PP4 catalytic subunit (1:500, Abcam ab115741), PP5 catalytic subunit[8] (1:500, gift of Dr S. Mochida, Kumamoto University, Japan), PP6 catalytic subunit (1:500, Bethyl A300-844A), poly-His (1:1000, Sigma A-7058) and GST (1:10,000, Sigma A-7340). Appropriate horseradish peroxidase-labeled secondary antibodies (Jackson Immunoresearch) were revealed by chemiluminescence (Pierce). All western blots are representative of at least three different experiments. Full scans of all western blots are provided as a Source Data file.

**Oocyte extracts, PP2A depletion, ammonium sulfate precipitation.** Metaphase II oocytes were homogenized in 10 volumes of Extraction Buffer (EB: 80 mM β-glycerophosphate pH 7.3, 20 mM EGTA, 15 mM MgCl₂) and centrifuged at 15,000 × g for 30 min at 4 °C. Extracts were then supplemented with 1 mM ATP, 100 mM MgCl₂ and 1 μM OA. Prophase oocytes were homogenized in 5 volumes of Purification Buffer (PB: 80 mM β-Glycerophosphate pH 7.4, 10 mM EDTA, 30 mM NaCl, 3 mM DTT) and centrifuged at 15,000 × g for 10 min at 4 °C. Extracts were then supplemented or not with 1 μM or 10 μM OA for 30 min at 18 °C, or with hexokinase (0.1 Units per μl) and 10 mM glucose, in the presence or in the absence of 20 mM phosphocreatine for 15 min at 18 °C. Recombinant PKI was added to the extracts at 0.15 mg/ml final concentration. For PP2A depletion, oocyte extracts were incubated twice for 30 min at 4 °C with either 50 μl of microcystin-Sepharose beads (Upstate) or 20 μl of magnetic GSH-beads, previously coupled or not to 2.5 μg of S67-thiophosphorylated GST-Arpp19. Supernatants were then used for further analysis after beads removal. For ammonium sulfate precipitation, proteins of extracts from 100 oocytes were precipitated by salting out using ammonium sulfate, successively 20% 40 and 60% as described[72]. Ammonium sulfate pellets were resuspended in 1 ml of PB.

**Cloning and recombinant protein purification.** Plasmids encoding either GST-tagged wild-type Arpp19[3], GST-S109A-mutated Arpp19[3], GST-p21Cip1[37], Histidine-tagged PKA catalytic subunit (a gift from Susan Taylor, Addgene plasmid #14921) and PKI[3] were previously described.

The cDNAs encoding either *Xenopus* catalytic PP1α and B55δ subunit were purchased from Thermo Fisher (ppp1ca 379914 Clone ID: 4682749 MXL1736-202771954) and GE healthcare (XCG ppp2cb cDNA cl5073873) and cloned into a pRN3 vector by PCR using primers encoding N-terminus Histidine-tag. The cDNA encoding the Cter-Arpp19 (amino-acids 68-117 of Arpp19) was cloned by PCR into a pGEX-6P-1 vector (See Supplementary Table 4). Recombinant proteins were produced in *E. coli* by autoinduction[73]. Wild-type, mutated and truncated GST-Arpp19 proteins, GST-p21Cip1, His-PKA catalytic subunit and PKI were purified as described in[3,37,74], dialyzed overnight against Phosphate Buffered Saline (PBS: 4.3 mM KH₂PO₄, 1.4 mM Na₂HPO₄ pH 7.4, 13.7 mM NaCl, 2.7 mM KCl) and stored at −80 °C.

**In vitro phosphorylation of Arpp19 and GST pull-down.** His-K71M-Gwl was obtained as described[24]. In vitro phosphorylation by PKA or thiophosphorylation by Gwl of recombinant GST-Arpp19 were respectively described in[26] and[24]. The double S67-S109 phosphorylated form of Arpp19 (pS67-pS109-Arpp19) and S67-phosphorylated form of S109A-Arpp19 (pS67-S109A-Arpp19) were produced by incubating 1 μg of either pS109-GST-Arpp19 or GST-Arpp19, previously coupled to 20 μl of magnetic GSH-beads, in 40 μl of metaphase II extracts for 2 h at 30 °C. After washing in PB, beads were stored at 4 °C for future use. In in vitro assays using purified phosphatases, S67-S109A-Arpp19 was phosphorylated in metaphase II extracts prior its elution from GSH-beads. For GST pull-down, GST-Arpp19 proteins were recovered from oocyte lysates as described previously[24].

**PP1 and B55δ purification.** mRNAs encoding Histidine-tagged catalytic PP1 and B55δ subunits were in vitro transcribed using mMESSAGE mMACHINE T3 Transcription Kit (Invitrogen, AM1348), in vitro polyadenylated using Poly(A) Tailing Kit (Invitrogen, AM1350) and purified using RNeasy Mini Kit (Qiagen, 74104). Prophase oocytes were injected with mRNAs, incubated for 18 h to allow mRNA expression and then homogenized in 10 volumes of Histidine Buffer (HB: 25 mM Hepes pH 7.2, 2 mM MgCl₂, 1 mM β-glycerophosphate). After centrifugation at 10,000 xg for 15 min at 4 °C, lysates from 5 oocytes expressing His-B55δ or 7.5 oocytes expressing His-PP1 were incubated with 40 μl of Co-beads for 2 h at 4 °C. After washing in HB, PP1 and PP2A-B55δ bound to Co-beads were either western blotted or assayed for phosphatase activities.

**S109-phosphatase, S67-phosphatase and PKA assays.** All assays used GST-Arpp19 or S109A-GST-Arpp19 at 1 μM final concentration.

Oocyte extracts, ammonium sulfate precipitates and fractions from chromatography columns: 1 μg of either pS109-GST-Arpp19 or GST-Arpp19, respectively used as substrates for S109-phosphatase or PKA, was coupled to 20 μl of magnetic GSH-beads for 30 min at 18 °C. After washing in PBS, GSH-beads coupled to Arpp19 proteins were incubated for 3 h at 30 °C with either 20 μl of oocyte extracts or 40 μl ammonium sulfate precipitates or 100 μl fractions from chromatography columns, supplemented with 1 mM ATP and 100 mM MgCl₂. pS109-GST-Arpp19 or GST-Arpp19 coupled to beads were then isolated and western blotted using antibodies directed against S109-phosphorylated Arpp19 and GST. In some experiments, doubly phosphorylated pS67-pS109-GST-Arpp19 was used as a substrate, under the same conditions as those described above, and western blotted using antibodies directed against either S109- or S67-phosphorylated Arpp19 and GST.

Purified PP1 and PP2A-B55δ: PP1 and PP2A-B55δ bound to Co-beads were incubated with 60 μl of HB supplemented with 100 mM MgCl₂ and substrates eluted from beads, either pS109-GST-Arpp19 or pS67-S109A-GST-Arpp19. At the indicated times, the reaction solution was centrifuged at 5000 × g for 5 s and 4 μl of supernatant were collected and analyzed by western blot using antibodies against either S109- or S67-phosphorylated Arpp19 or GST.

**Quantifications and statistics.** All western blot signals were quantified using Image J software. To quantify the endogenous level of Arpp19 phosphorylation in in vivo experiments, the S109 phosphorylation signal was divided by the corresponding total endogenous Arpp19 signal (pS109/$^{total}$Arpp19) for each condition. pS109/$^{total}$Arpp19 ratios were then normalized to the ratio of prophase oocytes.

In assays using recombinant GST-Arpp19, S109 phosphorylation signal was divided by its corresponding GST signal (pS109/$^{gst}$Arpp19) for each sample. In the phosphatase biochemical isolation and the time-course experiments, pS109/$^{gst}$Arpp19 ratios were normalized to pS109/$^{gst}$Arpp19 ratios of either the starting substrate or the time "0" respectively. For the other in vitro phosphatase assays, pS109/$^{gst}$Arpp19 ratios were normalized to the main value of pS109/$^{gst}$Arpp19 ratios of all biological replicates in order to compare them with each other. Each biological replicate was then plotted as an individual dot together with their mean and SEM values.

The graphs were prepared using the software Prism 8.

**Biochemical isolation of S109-phosphatase.** In total, 20,000 prophase oocytes were homogenized at 4 °C in 5 volumes of PB and successively centrifuged at 4 °C at 300 × g for 5 min, at 1100 × g for 20 min, at 15,000 × g for 20 min and at 100,000 × g for 2 h (rotor TFT-45). The supernatant was then incubated with recombinant PKI at 0.15 mg/ml final concentration for 30 min at 30 °C and then loaded on a Uno Q-25 column previously equilibrated in PB. Proteins were eluted with a linear gradient of 0 to 1 M NaCl. The flow through containing S109-phosphatase activity was loaded on a Mono Q 4.6/100PE column equilibrated in PB. Proteins were eluted with a linear 0 to 0.6 M NaCl gradient. Active S109-phosphatase fractions were dialyzed overnight against Dialysis Buffer (DB: 20 mM Hepes pH 7.4, 1 mM EDTA and 1 mM DTT) and then supplied with 1 M (NH₄)₂SO₄. Fractions were loaded on a Phenyl-Superose HR5/5 column equilibrated in DB supplemented with 1 M (NH₄)₂SO₄. Proteins were eluted with a linear 1 to 0 M (NH₄)₂SO₄ gradient. The active S109-phosphatase fractions were then lyophilized, resuspended in 200 μl of DB and further loaded on a gel size exclusion Superose 12 HR10/30 column.

**LC-MS/MS data acquisition and processing**

*Sample preparation.* Proteins were digested overnight at 37 °C with trypsin (Promega) in a 25 mM NH₄HCO₃ buffer (0.2 μg trypsin in 20 μl). The resulting peptides were desalted using ZipTip μ-C18 Pipette Tips (Pierce Biotechnology).

*Data acquisition.* Samples were analyzed using an Orbitrap Q-Exactive Plus coupled to a Nano-LC Proxeon 1000 equipped with an easy spray ion source (Thermo Scientific). On the Q-Exactive Plus instrument, peptides were loaded with an online preconcentration method and separated by chromatography using a Pepmap-RSLC C18 column (0.75 × 500 mm, 2 μm, 100 Å, from Thermo Scientific), equilibrated at 50 °C and operated at a flow rate of 300 nl/min. Peptides were eluted by a gradient of solvent A (H₂O, 0.1 % formic acid) and solvent B (100% acetonitrile, 0.1% formic acid). The column was first equilibrated 5 min with 95% of A, then B was raised to 35% in 93 min and finally, the column was washed with 80%B during 10 min and re-equilibrated at 95% A during 10 min. Peptides masses were analyzed in the Orbitrap cell in full ion scan mode at a resolution of 70,000 with a mass range of m/z 375-1500 and an AGC target of 3.106. MS/MS were performed in a Top 20 DDA mode. Peptides were selected for fragmentation by Higher-energy C-trap Dissociation (HCD) with a Normalized Collisional Energy of 27%, and a dynamic exclusion of 30 s. Fragment masses were measured in the Orbitrap cell at a resolution of 17,500, with an AGC target of 2.105. Monocharged peptides and unassigned charge states were excluded from the MS/MS acquisition. The maximum ion accumulation times were set to 50 msec for MS and 45 msec for MS/MS acquisitions respectively.

*Data processing.* The raw data were processed on Proteome Discoverer 2.2 or 2.4 with the mascot node (Mascot version 2.5.1) with the non-redundant protein database for *Xenopus laevis* taxonomy with a maximum of 2 missed cleavage sites. Precursor and fragment mass tolerance were set to 6 ppm and 0.02 Da respectively for Q-exactive Plus, 7 ppm and 0.5 Da respectively for Fusion. The following post-translational modifications were searched: Acetyl (Protein N-term), Oxidation (M), Phosphorylation (STY). The spectra were filtered using a 1% FDR with percolator node. For peptide identification, the data were searched against a *Xenopus laevis* database (July 2016, 17 742 entries) extracted with the NCBI protein search engine. These sequences come from various sources (UniprotKB/Swissprot, EMBL, RefSeq, GenBank, PDB, DDBJ and PIR).

**Reporting summary.** Further information on research design is available in the Nature Research Reporting Summary linked to this article.

## Data availability
All data supporting the findings of this study are available within the paper and its supplementary information file. The mass spectrometry proteomics data that support the findings of this study have been deposited on the ProteomeXchange Consortium via the PRIDE partner repository with the dataset identifier PXD022739. Other relevant data are available from the corresponding author upon reasonable request. Source data are provided with this paper.

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

## Acknowledgements
We thank all members of our laboratory, Dr O. Haccard and Dr T. Lorca for helpful discussions, as well as Dr T. Hunt and Dr E. Houliston for their precious advice. We are very grateful to Dr S. Mochida for providing the antibodies against B55δ, B56ε and PP5, as well as to Dr M. Bollen for the kind gift of the anti-PP1 antibody, Dr A. Nairn for the anti-Arpp19 antibody and Dr S. Taylor for the His-PKA plasmid. We thank the members of proteomics core facility at the Institute Jacques Monod for the LC-MS/MS experiments. T.L. received a PhD grant from Sorbonne University. This work was supported by The National Center for Scientific Research (CNRS), Sorbonne University, Paris University, the National Research Agency (ANR grants 13-BSV2-0008-01 and 18-CE13-0013-01 to CJ and AD respectively), the region Ile-de-France (SESAME, for funding parts of the LC-MS/MS equipment) and the ARC foundation (Association pour la Recherche sur le Cancer, grant PJA 2017-12-06-185 to AD and PDF22019120001043 to EMD).

## Author contributions
A.D. and C.J. contributed equally to this work. A.D. and C.J. conceived the original idea. T.L., E.M.D., A.D., and C.J. designed and planned the experiments, analyzed the data and wrote the paper. T.L., E.M.D., A.D., R.P., T.rL., M.M., and L.L. performed experiments and analyzed the results. R.P., T.rL., M.M., and L.L. assisted in writing the manuscript.

## Competing interests
The authors declare no competing interests.
