## [Peer Review File · Nature Communications]

Reviewers' comments:

Reviewer #1 (Remarks to the Author):

Cell cycle regulation by the PP2A-B55 phosphatase has attracted a lot of attention recently. It has become clear that PP2A-B55 is one of the key phosphatases opposing entry into mitosis in vertebrate cells. For efficient entry into mitosis PP2A-B55 therefore has to be inhibited, and this is achieved through competitive inhibition by the related, small, unstructured proteins Arpp19 and ENSA that bind to the PP2A-B55 catalytic site. Arpp19 and ENSA are only PP2A-B55 inhibitors in their phosphorylated form, and the Greatwall/MASTL kinase is required to carry out the key phosphorylations on Arpp19 and ENSA (S67 on Arpp19 or ENSA, respectively). The phosphorylated residue binds into the catalytic site of PP2A-B55 and is very slowly turned over by PP2A-B55 itself.

In vertebrate meiosis, Arpp19 has a second, important role in mediating prophase arrest. This involves phosphorylation by a distinct kinase, PKA, on a different residue of Arpp19, S109, and the effector of this form of regulation is currently not known. However, it is clear that Arpp19 S109 has to be dephosphorylated in order to release the prophase block upon progesterone exposure. The phosphatase carrying out this step has not been identified yet.

In this manuscript Lemonnier and colleagues set out to identify the *Xenopus* Arpp19-pS109 phosphatase using a biochemical strategy involving dephosphorylation assays with *Xenopus* whole extract and chromatographically separated fractions. On the whole the experiments are carefully conducted and the results look clean and convincing. The manuscript is a timely addition to the work that has been published on mitotic roles of Arpp19 and PP2A-B55 and should interest a wide readership. It would be a nice addition to the manuscript, though, to have an *in vitro* dephosphorylation assay with the recombinant pS109-Arpp19 substrate that is used throughout the paper and purified PP2A-B55, ideally in comparison to purified PP1, to show that the dephosphorylation results with extract can be recapitulated with purified proteins.

Minor comment: It would be very helpful for the reader to have the flow chart in Supplemental Figure 2 in the main figure since the description of the various chromatographic steps is somewhat confusing.

Reviewer #2 (Remarks to the Author):

See below

In the manuscript "The M-phase regulatory phosphatase PP2A-B55 δ opposes protein kinase A on Arpp19 to initiate meiotic division" by Lemonnier et al., the authors discover PP2A-B55 as the main phosphatase that dephosphorylates Arpp19 at the PKA site Ser109 in *Xenopus* meiosis. Previous work from the same lab showed that the phosphorylation of Arpp19 at Ser109 by PKA suppresses resumption of meiosis by a still unknown mechanism. This inhibitory phosphorylation is partially removed in response to hormones that can induce resumption of meiosis, e.g. progesterone. Thus, the identification of the responsible phosphatase is important to further understand the molecular

regulation of meiosis in vertebrates. Here, the authors use a combination of small molecules and phosphatase assays in fractionated oocyte extracts to reveal that PP2A-B55 is the main phosphatase responsible for dephosphorylation of Ser109 of Arpp19 in *Xenopus* meiosis. They further show, that the activity of PP2A-B55 against Arpp19 Ser109 is not regulated between Prophase I arrest and Greatwall activation and therefore conclude that the observed difference in Arpp19 Ser109 phosphorylation is due to altered PKA activity levels in response to progesterone.

The results presented in this manuscript are in principle interesting and important for the cell cycle community. The experiments mostly support the conclusions that are made. I recommend publication of this manuscript in *Nature Communications* after addressing the following points:

- page 3, introduction: MPF is explained as equal to Cdk1-CycB. However, it has been shown that Gwl also contributes to MPF activity (Hara et al., 2012). As Gwl also plays an important role in this manuscript, this should not be neglected.

- page 3, introduction: it is mentioned that PP2A-B55 δ dephosphorylates Myt1 and Cdc25 and Mochida et al., 2009 is given as reference. This reviewer could not find direct evidence for this reference, although it is mentioned in the discussion that this might be the case.

- page 4, introduction: this reviewer does not fully understand the meaning of the sentence "Importantly, the Gwl/Arpp19/PP2A-B55 module is under the control of Cdk1, being included inside the auto-amplification loop".

- in general, it is good that the immunoblots have been quantified, because the differences in the pS109 Arpp19 signal are sometimes hard to spot. Nevertheless, I think that it would strengthen the manuscript if for all experiments the mean with SD of the respective independent replicates would be shown and not the quantification of the shown single experiments. I also could not find a description of the quantification and normalization procedure in the methods section.

- Fig. 1A: are there estimates of how much exogenous ATP has been added in relation to the endogenous ATP? Have the authors tested if the overexpression of PKA has the same effect than increasing the PKA levels?

- Fig. 1A and 1C: Lane 2+3 in Fig.1A and Lane 1+2 in Fig.1C show the same treatments (Prophase Extract +/- PKI). However, the resulting phosphorylation of Ser109 Arpp19 seems to be very different.

The pS109 signal without PKI is very high in Fig.1C and almost absent in Fig.1A. From the Figure legend it seems that the treatments have been identical, i.e. without exogenous ATP. Could the authors please comment on this.

- Fig. 1A and 1B: In figure 1A, the authors start with pre-phosphorylated Arpp19. Under these conditions, PKA is not capable of efficiently maintaining the phosphorylation state of prephosphorylated

Arpp19. In contrast, according to Fig. 1B, PKA can efficiently de novo phosphorylate Arpp19 despite the presence of active B55. This reviewer is puzzled by this result.

- the authors took great efforts with their fractionation experiments to identify the Ser109 Arpp19 phosphatase. However, this reviewer found it hard to follow the experiments from the description in the text and the fact that the experiments are stretched over 3 figures and 1 table (+ the supplementary figures). This reviewer strongly suggests to move most – if not all – of the purification figure to the supplementary data, i.e. Fig. 2-5 and table 1.

- the three tables with the MS quantifications show slightly different amounts of proteins/accession numbers analyzed, e.g. B55beta is only in Exp1. Can the authors comment why this is the case.

- Fig. 6A: Arpp19-pS67-pS109 is dephosphorylated at both sites in prophase I extracts. The authors

conclude that this happens via PP2A-B55 in both cases. In theory (Williams et al., 2014), the presence

of pS67 should exclude the dephosphorylation of all other substrates, including pS109 (otherwise pS109 Arpp19 would also inhibit PP2A-B55 by unfair competition), until pS67 is dephosphorylated below the concentration of PP2A-B55. How do the authors explain that pS109 can be dephosphorylated in the presence of pS67? What are the respective kinetics of dephosphorylation? This is a very important point, which has to be clarified.

- Fig. 6A and 6B: According to 6B, the two bands are detected by the pS67 Ab. According to 6A, only the upper band is detected by the pS109 Ab. This indicates that S109 phosphorylation, but not 67 phosphorylation, retards the mobility of Arpp19 and, importantly, that the authors work with a mixed population of Arpp19 partly phosphorylated at S109. This is an important point, which has to be addressed.

- Fig. 6C: the μ Cys almost completely depletes PP2A-C from the extract (better than tpS67), but on the beads there is much less PP2A-C detectable for the μ Cys. Could the authors comment on this.

- Fig. 7B. The quantification does not really match the pS109 signal of the WB, see this reviewer's general comment on statistics.

- Fig. 8A: in the Pg+Cip condition, the difference in pS109 levels between 30' and 60' looks stronger than what is quantified. This might be partially due to the total level of Arpp19 present in the respective oocytes, but this figure would profit from the quantification of additional experiments.

- p. 18: "our present results reveal that PP2A-B55 also positively regulates meiosis resumption by" Which figure in the current ms does support this conclusion?

- p. 22, discussion: it is mentioned that Arpp19 contains the bipartite B55 recognition motif. This is a very strong argument, which can be easily tested and should be experimentally verified/falsified. In fact, this reviewer is puzzled that this experiment has not been done. This is an important point.

- p. 22, discussion: it is mentioned that "...PP2A-B55 δ ...is inhibited by Arpp19 phosphorylated at Ser67, a property not shared by the three other PP2A-B55 holoenzymes." This reviewer could not find the evidence for this in the listed references. Could the authors clarify this point.

- p. 22, discussion: Wee1/Myt1 are listed as PKA substrates. The reference listed investigates this in mouse and shows phosphorylation of Wee1B. To this reviewer's knowledge Wee1B should not be expressed in *Xenopus* prophase I arrested oocytes. To my knowledge, for Myt1 it is unclear if it is a PKA target. Could the authors comment on this?

- p. 23, discussion: the authors speculate that pS109 Arpp19 might be a PP1 inhibitor. This has been tested by Mochida et al., 2010 and it seems not to be the case.

- Overall, it seems that PP2A-B56 (as mentioned by authors) or other PPTases significantly contribute to pS109 Arpp19 dephosphorylation. Could the authors comment on the following points:

1) the profile of S109 phosphatase activity (mainly in fractions 5-8) in Fig.3B does not fit completely to the elution profile of PP2A-B55 δ (mainly in fraction 4-6).

2) Fig4A: there is much less B55 δ in fraction 12 than fraction 10, however for the S109 phosphatase activity the opposite is true. This is similar for Supplementary Figure 3B lanes 7-9.

3) Fig 5B: the activity seems to correlate exclusively with B56 ϵ (levels also high in MS analysis of the corresponding fraction).

4) Fig. 6C+D: The μ Cys and tpS67 precipitate similar amounts of B55, however the pS109 phosphatase activity is much higher with the μ Cys, which suggests that another PP2A holoenzyme contributes significantly. Can the authors provide a B56 blot for the experiment shown in Fig. 6C?

5) the appearance of PP2A-B56 in the three different MS experiments is somewhat inconsistent. Can the authors explain where it went in experiment 3, where it is still present in the input.

Article title: *The M-phase regulatory phosphatase PP2A-B55 δ opposes protein kinase A on Arpp19 to initiate meiotic division.* Lemonnier *et al.*

Revised version

Answers to reviewers

Reviewer #1 (Remarks to the Author):

Cell cycle regulation by the PP2A-B55 phosphatase has attracted a lot of attention recently. It has become clear that PP2A-B55 is one of the key phosphatases opposing entry into mitosis in vertebrate cells. For efficient entry into mitosis PP2A-B55 therefore has to be inhibited, and this is achieved through competitive inhibition by the related, small, unstructured proteins Arpp19 and ENSA that bind to the PP2A-B55 catalytic site. Arpp19 and ENSA are only PP2A-B55 inhibitors in their phosphorylated form, and the Greatwall/MASTL kinase is required to carry out the key phosphorylations on Arpp19 and ENSA (S67 on Arpp19 or ENSA, respectively). The phosphorylated residue binds into the catalytic site of PP2A-B55 and is very slowly turned over by PP2A-B55 itself.

In vertebrate meiosis, Arpp19 has a second, important role in mediating prophase arrest. This involves phosphorylation by a distinct kinase, PKA, on a different residue of Arpp19, S109, and the effector of this form of regulation is currently not known. However, it is clear that Arpp19 S109 has to be dephosphorylated in order to release the prophase block upon progesterone exposure. The phosphatase carrying out this step has not been identified yet.

In this manuscript Lemonnier and colleagues set out to identify the *Xenopus* Arpp19-pS109 phosphatase using a biochemical strategy involving dephosphorylation assays with *Xenopus* whole extract and chromatographically separated fractions. On the whole the experiments are carefully conducted and the results look clean and convincing. The manuscript is a timely addition to the work that has been published on mitotic roles of Arpp19 and PP2A-B55 and should interest a wide readership.

Answers to reviewer 1

We would like to thank the reviewer for his/her positive feedback on our article and for the suggestions for improvement.

1. Reviewer: *It would be a nice addition to the manuscript, though, to have an *in vitro* dephosphorylation assay with the recombinant pS109-Arpp19 substrate that is used throughout the paper and purified PP2A-B55, ideally in comparison to purified PP1, to show that the dephosphorylation results with extract can be recapitulated with purified proteins.*

Authors: The experience suggested by the reviewer is indeed a good guarantee of the result obtained by our purification procedure. To address this issue, the most difficult point was to overcome the technical obstacle of purifying the trimeric PP2A-B55 (PP1 is commercially available and can *in vitro* operate without any regulatory subunit). We have studied the protocols described in the literature, in particular those using Arpp19 as a substrate. Mochida *et al.*¹ describe a protocol based on the production of *Xenopus* GST-PP2A-A in bacteria, and *Xenopus* PP2A-C and rat His-B55 δ using baculovirus expression systems. After mixing the three proteins, a trimeric complex formed of *Xenopus* PP2A-A/C and rat B55 δ is purified on glutathione beads and then on Nickel beads. We did not have access to a cellular system allowing baculovirus expression, and were reluctant to use a heterologous rat-*Xenopus* holoenzyme. In addition, we wanted to compare PP2A-B55 and PP1 activities produced with comparably purified enzymes. We therefore adopted the following strategy/protocol: (1) *in vitro* translation of mRNAs coding *Xenopus* His-B55 δ and *Xenopus* His-PP1 (catalytic subunit); (2) mRNA injection in prophase oocytes to express both proteins and to allow B55 δ binding to endogenous PP2A-C and PP2A-A; (3) purification of His-PP1 and His-B55 δ from the injected oocytes using Cobalt-beads. Dephosphorylation assays of Arpp19 phosphorylated either at S109 or S67 were then performed using

these purified enzymes. Results are illustrated in new Fig. 6. PP1 is unable to dephosphorylate Arpp19 at S67 and weakly dephosphorylates S109. PP2A-B55 δ is efficient on both residues.

2. Reviewer: *Minor comment: It would be very helpful for the reader to have the flow chart in Supplemental Figure 2 in the main figure since the description of the various chromatographic steps is somewhat confusing.*

Authors: Both reviewers found the description of the various chromatographic steps somewhat hard to follow. It is clear that we have not been able to present in a sufficiently clear manner this heavy series of experiments, which are central to the article. We thank the reviewer for his/her suggestion, which we adopted (new Fig. 3a).

Reviewer #2 (Remarks to the Author):

In the manuscript "The M-phase regulatory phosphatase PP2A-B55 δ opposes protein kinase A on Arpp19 to initiate meiotic division" by Lemonnier et al., the authors discover PP2A-B55 as the main phosphatase that dephosphorylates Arpp19 at the PKA site Ser109 in *Xenopus* meiosis. Previous work from the same lab showed that the phosphorylation of Arpp19 at Ser109 by PKA suppresses resumption of meiosis by a still unknown mechanism. This inhibitory phosphorylation is partially removed in response to hormones that can induce resumption of meiosis, e.g. progesterone. Thus, the identification of the responsible phosphatase is important to further understand the molecular regulation of meiosis in vertebrates. Here, the authors use a combination of small molecules and phosphatase assays in fractionated oocyte extracts to reveal that PP2A-B55 is the main phosphatase responsible for dephosphorylation of Ser109 of Arpp19 in *Xenopus* meiosis. They further show, that the activity of PP2A-B55 against Arpp19 Ser109 is not regulated between Prophase I arrest and Greatwall activation and therefore conclude that the observed difference in Arpp19 Ser109 phosphorylation is due to altered PKA activity levels in response to progesterone.

The results presented in this manuscript are in principle interesting and important for the cell cycle community. The experiments mostly support the conclusions that are made. I recommend publication of this manuscript in Nature Communications after addressing the following points:

Answers to reviewer 2

We would like to thank the reviewer for his/her positive feedback on our article and for the suggestions for improvement.

1. Reviewer: *page 3, introduction: MPF is explained as equal to Cdk1-CycB. However, it has been shown that Gwl also contributes to MPF activity (Hara et al., 2012). As Gwl also plays an important role in this manuscript, this should not be neglected.*

Authors: In agreement with the reviewer, we have modified the text and added the appropriate reference² (page 3, lines 54).

2. Reviewer: *page 3, introduction: it is mentioned that PP2A-B55 δ dephosphorylates Myt1 and Cdc25 and Mochida et al., 2009 is given as reference. This reviewer could not find direct evidence for this reference, although it is mentioned in the discussion that this might be the case.*

Authors: Given that (1) Cdc25 and Wee1/Myt1 are direct substrates of Cdk1, (2) they are dephosphorylated by PP2A, and (3) the specific PP2A-B55 isoform dephosphorylates Cdk1 substrates, it is often proposed in the literature that PP2A-B55 isoform directly dephosphorylates Cdc25 and Wee1/Myt1^{1, 3, 4}. However only few papers show by biochemical ways or mathematical models that this is really the case^{5, 6, 7}. We agree with the reviewer that our wording was improper and we have changed the phrase and the references (page 3, lines 67).

3. Reviewer: page 4, introduction: this reviewer does not fully understand the meaning of the sentence "Importantly, the Gwl/Arpp19/PP2A-B55 module is under the control of Cdk1, being included inside the auto-amplification loop".

Authors: Gwl activation and hence the subsequent PP2A-B55 δ inhibition require Cdk1 activity, either achieved by Cdk1-Cyclin B in *Xenopus* oocytes and extracts^{8, 9, 10} and Cdk1-Cyclin A in human cells¹¹. Being dependent on Cdk1 activity, the Gwl/Arpp19/PP2A-B55 module does not act as a trigger to launch Cdk1 activation, but is part of the auto-amplification loop where Cdk1 not only phosphorylates its own activation module, Cdc25-Myt1, but also indirectly inactivates its antagonizing protein phosphatase, PP2A-B55 δ , through Gwl-Arpp19. We agree with the reviewer that our wording was too elusive and we have changed the phrase and the references (page 4, lines 98-99).

4. Reviewer: in general, it is good that the immunoblots have been quantified, because the differences in the pS109 Arpp19 signal are sometimes hard to spot. Nevertheless, I think that it would strengthen the manuscript if for all experiments the mean with SD of the respective independent replicates would be shown and not the quantification of the shown single experiments. I also could not find a description of the quantification and normalization procedure in the methods section.

Authors: As requested by the reviewer, all the *in vitro* experiments are now illustrated by one representative experiment together with the mean values and SEM of independent replicates and experiments. Accordingly, the following figures have been changed (new numbering): Fig. 1b, d and f; Fig. 2c and e; Fig. 6c and e; Fig. 7b, d, f and h; Fig. 9e and f; Sup. Fig. 2b.

We apologize for having described our quantification method in a far too brief manner. This is now the subject of a paragraph in the "Methods" section (page 20, lines 593-607).

Regarding *in vivo* experiments (Fig. 8 and Fig. 9a-b), mean values with SEM are not scientifically relevant due to the high level of variability in the timing of Cdk1 activation. Depending on the females, it occurs between 2h and 6h. Among oocytes from a single female, GVBD occurs over an interval of 2h. Moreover, Arpp19 dephosphorylation at S109 is a transient event starting 15 min to 1.5h after progesterone stimulation depending on the females. Therefore, in contrast to *in vitro* experiments where molecular events occur in a synchronized manner and are identical whatever the female donor, thus allowing quantification and mean values with SEM, *in vivo* data, although reproducible, cannot be summed up by mean values that erase significant molecular fluctuations, due to the variability of their timing. It is why they are illustrated by a representative experiment.

5. Reviewer: Fig. 1A: are there estimates of how much exogenous ATP has been added in relation to the endogenous ATP? Have the authors tested if the overexpression of PKA has the same effect than increasing the PKA levels?

Authors: The concentration of ATP is 0.6 to 1mM in the intact *Xenopus* oocyte^{12, 13}. During lysis of oocytes, the endogenous levels of ATP strongly decrease due to the activity of ATPases released by intracellular compartment lysis. We added 1 mM exogenous ATP, in order to generate equivalent extracts with a saturated level of ATP. To circumvent the variability of the ATP concentration in extracts, a number of laboratories working with *Xenopus* egg extracts have used an ATP-regenerating system, which allows to maintain a stable concentration of ATP¹⁴. Given the remark of the reviewer and to use extracts with a stable and identical ATP level, we repeated the experiments with 3 different females and two additional conditions: (1) hexokinase and glucose to fully deplete ATP and provide extracts devoid of endogenous ATP and (2) phosphocreatine, which regenerates ATP and ensures a high and stable level of ATP (there is no need to add creatine kinase since it is expressed in the oocyte¹⁵). These new experiments are illustrated in new Fig. 1a-d. The results are identical as previously. In the absence of ATP (hexokinase + glucose), Arpp19 is efficiently dephosphorylated in extracts (Fig. 1a-b) and cannot be phosphorylated by PKA (Fig. 1c-d). In contrast, in the presence of a constant level of ATP (hexokinase + glucose + phosphocreatine), Arpp19 dephosphorylation does not occur unless PKA is inhibited by PKI (Fig. 1a-b) and is efficiently phosphorylated by PKA (Fig. 1c-d).

We had tested the addition of PKA-C. In the absence of ATP, it does not modify S109 dephosphorylation, as expected since the kinase cannot work without ATP. In the presence of ATP, the addition of the C catalytic subunit of PKA in extracts does not produce any remarkable changes in the phosphorylation level of Arpp19, probably because the endogenous activity of PKA is already at its maximum level. In contrast, PKA inhibition by PKI fully abolishes Arpp19 phosphorylation at S109, thus demonstrating that PKA is the specific kinase phosphorylating Arpp19 at S109.

6. Reviewer: *Fig. 1A and 1C: Lane 2+3 in Fig.1A and Lane 1+2 in Fig.1C show the same treatments (Prophase Extract +/- PKI). However, the resulting phosphorylation of Ser109 Arpp19 seems to be very different. The pS109 signal without PKI is very high in Fig.1C and almost absent in Fig.1A. From the Figure legend it seems that the treatments have been identical, i.e. without exogenous ATP. Could the authors please comment on this.*

Authors: The experiment illustrated in former Fig. 1c was performed in the presence of ATP. Hence, lanes 1+2 of Fig. 1c had to be compared to lanes 4+5 of former Fig. 1a (and not lanes 2+3) and both results are consistent. We apologize for having omitted this ATP clarification in the legend. Since the "Methods" section mentioned that the buffer used in both S109-phosphatase and PKA assays contains ATP, we did not re-state this point in the legends and only mention conditions where ATP has been omitted (former Fig. 1a-b). We corrected the legends. Independently of this remark, Fig. 1c has been changed to answer to point 4 regarding statistics (and is now Fig. 1e-f).

7. Reviewer: *Fig. 1A and 1B: In figure 1A, the authors start with pre-phosphorylated Arpp19. Under these conditions, PKA is not capable of efficiently maintaining the phosphorylation state of prephosphorylated Arpp19. In contrast, according to Fig. 1B, PKA can efficiently de novo phosphorylate Arpp19 despite the presence of active B55. This reviewer is puzzled by this result.*

Authors: In former Fig. 1a, PKA is indeed not capable of efficiently maintaining the phosphorylation state of prephosphorylated Arpp19 in the absence of ATP, since the kinase cannot catalyze any enzymatic reaction in the absence of ATP. As a consequence, it does not antagonize S109-phosphatase that dephosphorylates Arpp19, and Arpp19 phosphorylation cannot be maintained since the kinase is inactive. In contrast, in the presence of ATP, the kinase can work and Arpp19 phosphorylation is maintained despite the presence of S109-phosphatase activity. This suggests that PKA is dominant over S109-phosphatase. This conclusion is confirmed by Fig. 1b: in the presence of S109-phosphatase and ATP (meaning PKA active), Arpp19 is phosphorylated, showing that PKA activity dominates the phosphatase. It should be noted that the extracts "without ATP" contain in fact low, but variable levels of endogenous ATP (see our answer to point 5), giving rise to more or less active PKA. This issue of variability has been overcome by the new experiences now illustrated in Fig. 1a-d: the hexokinase efficiently eliminates endogenous ATP from extracts and the ATP-regenerating system ensures an identical and constant ATP level whatever the extract. Under these conditions, the results are identical to the previous ones: both PKA and S109-phosphatase are active in prophase, with PKA dominating.

8. Reviewer: *the authors took great efforts with their fractionation experiments to identify the Ser109 Arpp19 phosphatase. However, this reviewer found it hard to follow the experiments from the description in the text and the fact that the experiments are stretched over 3 figures and 1 table (+ the supplementary figures). This reviewer strongly suggests to move most – if not all – of the purification figure to the supplementary data, i.e. Fig. 2-5 and table 1.*

Authors: Both reviewers found the description of the various chromatographic steps somewhat hard to follow. We have not been able to present in a sufficiently clear manner this heavy series of experiments, which are central to the article. The two reviewers gave us rather opposite advice to address this issue. The other reviewer suggests moving the flow chart in the former Supplementary Figure 2 to the main figure whereas this reviewer suggests to move the purification figure to the Supplementary data. We favored the option of the other reviewer. The purification experiment is indeed the basis of the article: it allowed to identify the specific phosphatase dephosphorylating

Arpp19 at S109, and all the other experiments of the article derive from this original one. Moreover, because of their central position in the article, these biochemical approaches, which are quite heavy and complex, must be reproducible by any other lab, and as such, must be described in a detailed and rigorous way. It is therefore particularly important that this purification experiment occupies a central position in the article, as it is the keystone of the article. We have therefore moved the scheme of the protocol, which was illustrated as a supplementary data, in Figure 3a, so that readers can follow the experiment more easily.

9. Reviewer: *the three tables with the MS quantifications show slightly different amounts of proteins/accession numbers analyzed, e.g. B55beta is only in Exp1. Can the authors comment why this is the case?*

Authors: We apologize for these errors, due to misuse of different libraries, some mixing between *laevis* and *tropicalis* species and the presence of various isoforms with different accession numbers. The bioinformatician in charge of these analyses has corrected all the errors. For a better understanding of the changes, we provide a table that summarizes the new LC-MS/MS analysis done for the three different purifications (see below). In this table, proteins whose description, accession numbers and/or gene symbols have been modified are written in red in orange boxes. The new analysis allows to replace all proteins identified in *X. tropicalis* by the *X. laevis* corresponding ones. In addition, proteins are often expressed under several distinct isoforms. The genome of *X. laevis* being pseudo-tetraploid, proteins are encoded by genes localized on either a single chromosome (indicated as “S” and “L” in the table) or the two homologous chromosomes. Regarding the different amounts of proteins recovered from one purification to another, this can be explained by the different amounts of starting material (10,000 oocytes for experiments 2 and 3; 20,000 oocytes for experiment 1) provided from 3 distinct females.

Tables Family	Description	Accession (gi numbers)			Gene Symbols
		Table 1	Suppl. Table 1a	Suppl. Table 1b	
PP2A-A	Phosphorylase phosphatase/PP2A-A	963085			
	Structural subunit A β	> 148230849	148230849	148230849	ppp2r1b; ppp2r1b.S
	65 kDa structural subunit A β isoform-like		148227844	148227844	LOC398563
	Structural subunit A α	148222150	148222150	148222150	ppp2r1a-b; ppp2r1a.S
PP2A-C	Structural subunit A α	148224496	148224496	148224496	ppp2r1a-a; ppp2r1a.L
	Catalytic subunit C α	148230509	148230509	148230509	ppp2ca; ppp2ca.L
B55	Catalytic subunit β [X. tropicalis]	53749698	53749698	53749698	
	> Catalytic subunit C β [X. laevis]	> 148234623	> 148234623	> 148234623	ppp2cb; ppp2cb.L
B56	Regulatory subunit B α	148234757	963087 > 148234757	148234757	ppp2r2a; ppp2r2a.S
	Regulatory subunit B δ	147900119	147900119	147900119	ppp2r2d; ppp2r2d.S
	Regulatory subunit B β	148231951			ppp2r2b; ppp2r2b.L
PP1	Regulatory subunit B' α	168693595	168693595	168693595	ppp2r5a; ppp2r5a.S
	Regulatory subunit B' β	148234627		148234627	ppp2r5b; ppp2r5b.L
	Regulatory subunit B' γ	148236119	148236119	148236119	ppp2r5c; ppp2r5c.L
	Regulatory subunit B' ϵ	148236023	148236023	148236023	ppp2r5e; ppp2r5e.S
PP1-R	Regulatory B' ϵ	147902694	147902694	147902694	ppp2r5e.L
	Catalytic subunit C α	147903539	147903539		ppp1ca; ppp1ca.L
	Catalytic subunit B γ	148224548		148224548	ppp1cc-B, ppp1cc.S
PP2C	Catalytic subunit β [X. tropicalis]	58332786			
	> Catalytic subunit C β [X. laevis]	> 148224494			ppp1cb.L
PP3	Regulatory subunit 11	148229062			ppp1r11; ppp1r11.L
	PP1A	147905165	147905165	147905165	ppm1a; ppm1a.L
PP4	Mg ²⁺ /Mn ²⁺ dependent, 1B	148227634	148227634	148227634	ppm1b; ppm1b.L
	Catalytic subunit C α	148235473	148235473 or 148235616	148235473 or 148235616	ppp3ca; ppp3ca.L ppp3ca.S
PP4-R	Catalytic subunit [X. tropicalis]	45360541	45360541		
	> Catalytic subunit [X. laevis]	> 148226861	> 148226861		ppp4c.S
PP6	Regulatory subunit 2-B			147901735	ppp4r2; ppp4r2.S
	Catalytic subunit	147906292	147906292	147906292	ppp6c; ppp6c.L

10. Reviewer: *Fig. 6A: Arpp19-pS67-pS109 is dephosphorylated at both sites in prophase I extracts. The authors conclude that this happens via PP2A-B55 in both cases. In theory¹⁶, the presence of pS67 should exclude the dephosphorylation of all other substrates, including pS109 (otherwise pS109 Arpp19 would also inhibit PP2A-B55 by unfair competition), until pS67 is dephosphorylated below the concentration of PP2A-B55. How do the authors explain that pS109 can be dephosphorylated in the presence of pS67?*

What are the respective kinetics of dephosphorylation? This is a very important point, which has to be clarified.

Authors: We have addressed the reviewer concern. The results of the previous Fig. 6a-b illustrated the phosphorylation levels of GST-Arpp19 after a 3 h incubation time. We performed new experiments by incubating the doubly-S67-S109 phosphorylated GST-Arpp19 from 0 to 3h in prophase extracts in which both Greatwall and PKA cannot function (no ATP, and Greatwall is inactive in prophase). The results of 8 experiments are now illustrated in the new Fig. 7a-d. Clearly, S67 is dephosphorylated much faster than S109: Arpp19 is dephosphorylated at S67 by 70% within 30 min whereas only 30% is dephosphorylated at S109 at that time. This is in agreement with the model in which S67-phosphorylated Arpp19 binds B55 tightly and is slowly dephosphorylated, maintaining PP2A-B55 inhibited. In our prophase extracts, Arpp19 is not dephosphorylated at S109 as long as the S67 residue remains phosphorylated. Moreover, S67, once dephosphorylated, cannot be rephosphorylated by Greatwall since this kinase is inactive. Hence, its inhibitory action towards PP2A-B55 is eventually lost, explaining why PP2A-B55 can then dephosphorylate Arpp19 at S109. At 3h (previous Fig. 6a-b, now Supplementary Fig. 3a-b), both residues are then dephosphorylated. The new Fig. 7a-d shows that their respective kinetics are strongly different, S109 dephosphorylation depending on prior S67 dephosphorylation that releases PP2A-B55 from its inhibitory action.

11. Reviewer: *Fig. 6A and 6B: According to 6B, the two bands are detected by the pS67 Ab. According to 6A, only the upper band is detected by the pS109 Ab. This indicates that S109 phosphorylation, but not 67 phosphorylation, retards the mobility of Arpp19 and, importantly, that the authors work with a mixed population of Arpp19 partly phosphorylated at S109. This is an important point, which has to be addressed.*

Authors: We have investigated the origin of the two bands mentioned by the referee. The lower band is recognized by the anti-phospho-S67 antibody and very often by the anti-GST antibody but never by the anti-phospho-S109 antibody (see Supplementary Fig. 2, Supplementary Fig. 3a-b, but also our previous articles^{9, 10, 17}). As proposed by the referee, the upper band could result from a mobility shift due to S109 phosphorylation. However, the constant recognition of both bands by the anti-GST antibody, independently of the phosphorylation levels at S109, does not support this hypothesis. We favored the hypothesis that the bacterially produced recombinant GST-Arpp19 protein could be partially proteolyzed during either its expression in bacteria or its purification. To test this idea, we analyzed by mass spectrometry the sequences of both bands as well as their phosphorylation level (see figure below). The results confirmed that the upper band corresponds to the full-length GST-Arpp19 protein, whereas the lower band corresponds to a GST-Arpp19 form lacking the 11 last amino acids at its C-terminal end. This form of GST-Arpp19 is therefore truncated and does not include anymore the S109 residue while it still contains S67 (it misses the C-ter KPS₁₀₉LVASKLAG peptide). As a consequence, the upper band, which corresponds to full-length GST-Arpp19, can be phosphorylated at both S67 and S109 and recognized by both anti-phospho-antibodies. In contrast, the lower band that is devoid of S109 is only recognized by the anti-phosphoS67-antibody but not by the anti-phospho-S109. Since both proteins contain the GST fragment, positioned in N-terminus of Arpp19, they are both recognized by the anti-GST antibody.

Figure. The upper band (FL-^{gst}Arpp19) and the lower band (T-^{gst}Arpp19) were cut out from a 10% polyacrylamide gel after Coomassie blue staining and analyzed by LC-MS/MS. The two upper panels show the peptides identified by LC-MS/MS analysis. The C-terminal peptide, PSLVASKLAG, is never recovered from T-^{gst}Arpp19. Additionally, when GST-Arpp19 has been phosphorylated at S109, phosphorylated S109 was detected by LC-MS/MS in the peptides generated by FL-^{gst}Arpp19 (residue in red) but never in the peptides generated by T-^{gst}Arpp19.

FL-9stArpp19

```

msgenqetkaqeessaleqkeiddkvvspekseeiklkarypnlgpkpggsdfllrkqlkgqkyfdsgdynmakakmknkqlptaasdktevtgdhiptpdlpqrkpslvasklag
[k].aqeessaleqk.[e]
[k].aqeessaleqkeiddk.[v]
[k].aqeessaleqkeiddkvvspek.[s]
[k].eiddkvvspek.[s]
[k].eiddkvvspekseeik.[l]
[k].vvspekseeik.[l]
[k].arypnlgpkpggsdfllr.[k]
[r].ypnigpkpggsdfllr.[r]
[r].ypnigpkpggsdfllr.[k]
[k].gqkyfdsgdynmak.[a]
[k].yfdsgdynmakak.[im]
[k].yfdsgdynmak.[a]
[k].qlptaasdk.[t]
[k].qlptaasdktevtgdhiptpdlpqr.[k]
[k].nkqlptaasdktevtgdhiptpdlpqr.[k]
[k].qlptaasdktevtgdhiptpdlpqrkpslvask.[l]
[k].tevtgdhiptpdlpqrkpslvask.[l]
[k].tevtgdhiptpdlpqr.[k]
[r].kpslvask.[l]

```

T-9stArpp19

```

msgenqetkaqeessaleqkeiddkvvspekseeiklkarypnlgpkpggsdfllrkqlkgqkyfdsgdynmakakmknkqlptaasdktevtgdhiptpdlpqrkpslvasklag
[k].aqeessaleqk.[e]
[k].aqeessaleqkeiddk.[v]
[k].aqeessaleqkeiddkvvspek.[s]
[k].eiddkvvspek.[s]
[k].eiddkvvspekseeik.[l]
[k].vvspekseeik.[l]
[k].vvspekseeikk.[a]
[k].arypnlgpkpggsdfllr.[k]
[r].ypnigpkpggsdfllr.[r]
[r].ypnigpkpggsdfllr.[k]
[k].yfdsgdynmakak.[im]
[k].yfdsgdynmak.[a]
[k].qlptaasdktevtgdhiptpdlpqr.[k]
[k].tevtgdhiptpdlpqr.[k]

```

Full length Arpp19

GST_MSGENQETKAQEESALEQKEIDDKVVSPEKSEEIKLKARYPNLGPKPGGSDFLLRRLQKGQKYFDS₆₇GDY
 NMAKAKMKNKQLPTAASDKTEVTGDHIPTPQDLPQRKPS₁₀₉LVASKLAG

The production of the lower band is variable between purification batches (and despite the fact that this purification is always performed at 4°C in the presence of anti-protease agents), which explains that the lower band is more or less present in the western blots probed with the anti-GST antibody. In summary, the analysis of experiences should only take into account the upper band. These new data are now available in Supplementary Fig. 1a-b.

12. Reviewer: Fig. 6C: the μ Cys almost completely depletes PP2A-C from the extract (better than tpS67), but on the beads there is much less PP2A-C detectable for the μ Cys. Could the authors comment on this?

Authors: We agree with the reviewer that in former Fig. 6c, PP2A-C is weakly detected on the μ Cys beads, compared to PP2A-A and B55, and compared to tpS67 beads. Given that PP2A-C should be in the same amount as PP2A-A (both are combined in a dimer and are not present under a free form), we assumed that the low detection of PP2A-C on μ Cys beads was a western blot issue. Hence, we used the same samples and loaded a new gel that was western blotted with the same antibodies (plus an anti-B56 antibody to answer another request of the referee, point 20.4, and an anti-karyopherin antibody as a negative control). The results are now illustrated in Fig. 7e-f and Supplementary Fig. 3c. Although PP2A-A/C is not fully depleted from the supernatant after an incubation with either μ Cys or

tpS67 beads, identical amounts of PP2A-A and PP2A-C are recovered on both types of beads. As expected, B55 δ is much more enriched on tpS67 beads compared to μ Cys beads.

13. Reviewer: *Fig. 7B. The quantification does not really match the pS109 signal of the WB, see this reviewer's general comment on statistics.*

Authors: All the quantifications of western blots have been done independently by three authors, one of them ignoring the nature and the order of the loaded samples. These blind quantifications provided strictly identical results. We are confident that quantification provides much more reliable data than visual perception.

14. Reviewer: *Fig. 8A: in the Pg+Cip condition, the difference in pS109 levels between 30' and 60' looks stronger than what is quantified. This might be partially due to the total level of Arpp19 present in the respective oocytes, but this figure would profit from the quantification of additional experiments.*

Authors: All western blots are representative of at least three different experiments. As explained above, all the quantifications of western blots have been done independently by three authors, one of them ignoring the nature and the order of the loaded samples. These blind quantifications provided identical results, whether within the same experiment or for all the experiments carried out. In all experiments, progesterone induces a drop in S109 phosphorylation level (at 30, 60 or 90 min depending on the females), plus or minus Cip1. In the presence of Cip1, Arpp19 stays under this S109-hypophosphorylated state whereas it is phosphorylated back at S109 without Cip1.

15. Reviewer: *p. 18: "our present results reveal that PP2A-B55 also positively regulates meiosis resumption by" Which figure in the current ms does support this conclusion?*

Authors: We agree with the reviewer that this sentence is more an interpretation than a factual presentation of the data, as should be the case in the "Results" section. The idea was that, as B55 allows the dephosphorylation of S109, which is needed for the resumption of meiosis, B55 plays a positive role. We changed that text, removing this overinterpretation (page 11, lines 336-339).

16. Reviewer: *p. 22, discussion: it is mentioned that Arpp19 contains the bipartite B55 recognition motif. This is a very strong argument, which can be easily tested and should be experimentally verified/falsified. In fact, this reviewer is puzzled that this experiment has not been done. This is an important point.*

Authors: Cundell *et al.*¹⁸ show that B55 substrates, including Ensa, share a bipartite polybasic recognition determinant (BPR). As human Ensa, *Xenopus* Arpp19 contains three basic patches making up the B55 BPR. As suggested by the referee, we designed a protein with alanine-scanning mutations at basic residues of these patches, what, after Cundell *et al.*, slows down the dephosphorylation kinetics of Ensa but does not prevent it. We expressed and purified the BPR mutant from bacteria and monitored its phosphorylation in oocyte extracts: metaphase II extracts for S67 phosphorylation and prophase extracts for S109 phosphorylation. As shown in the figure below, the BPR mutant behaves as wild-type Arpp19 in terms of S109 phosphorylation: it is phosphorylated at S109 and inhibiting PKA with PKI promotes its dephosphorylation at S109. Similarly, the BPR mutant is phosphorylated at S67 as the wild-type protein. In rare cases, the extent of S67 phosphorylation of the BPR mutant was reduced in comparison to wild-type Arpp19, as shown in the figure, in strong contrast to the results of Cundell *et al.* (where S67 dephosphorylation is slowed down). We did not observe any significant difference in the binding of BPR and wild-type proteins to PP2A-A/C in oocyte extracts.

We injected both proteins in *Xenopus* oocytes. We could not detect significant differences between the wild-type and the mutant on the kinetics of maturation induced by progesterone. In rare cases, the positive effect of wild-type Arpp19 on meiosis resumption was not reproduced by the BPR mutant (see figure below), maybe in correlation with the reduced S67 phosphorylation level observed in few cases in the extracts.

Wild-type Arpp19 – The three basic patches making up the B55 BPR are framed.

MSGENQETKAQEESSALEQKEIDDKVVSPEKSEET**KLKAR**YPNLGPKPGGSDFL**KR**LQKGQKYFDS₆₇GDYNMA
KAKMKNKQLPTAASDKTEVTGDHIPTPQDLPQRKPS₁₀₉LVASKLAG

BPR Arpp19 mutant – Alanine mutations in the three basic patches making up the B55 BPR are indicated in red.

MSGENQETKAQEESSALEQKEIDDKVVSPEKSEET**ALAA**YPNLGPKPGGSDFL**AA**LQKGQKYFDS₆₇GDYNMA
AAAMANAQLPTAASDKTEVTGDHIPTPQDLPQRKPS₁₀₉LVASKLAG

Figure. BPR does not affect significantly dephosphorylation of *Xenopus* Arpp19.

(a-b) Wild-type-GST-Arpp19 (WT) or BPR-GST-Arpp19 (BPR) coupled to GSH beads were incubated either in prophase (a) or in metaphase II (b) extracts in the presence of ATP. After 30 min incubation, prophase extracts were supplied with PKI for 2.5 h. The beads were then isolated and S109 and S67 phosphorylations were analyzed by western blot using phospho-S109-Arpp19, phospho-S67-Arpp19 and GST antibodies. **(a)** S109 phosphorylation (pS109) in prophase extracts supplemented or not with PKI. **(b)** S67 phosphorylation (pS67) in metaphase II extracts. **(c)** GVBD time-course. Prophase oocytes were injected with 300 ng of either WT-GST-Arpp19 (WT) or BPR-GST-Arpp19 (BPR) and then stimulated with progesterone (Pg) 18h later.

This shows that, in *Xenopus* oocytes, the BPR mutant does not act differently from the wild-type protein. This apparent discrepancy with Cundell's results can be explained by the difference of cellular systems (Human cell lines versus *Xenopus* oocytes). It is important to note that in Cundell's paper, the BPR mutant only slows down the action of B55 but does not prevent it from dephosphorylating its substrate. In our hands, in rare cases, it is even the opposite situation, with PP2A being more active at S67 with the mutant (maybe because the mutant is a less efficient inhibitor of PP2A due to a weaker binding). B55 probably interacts normally with endogenous Arpp19 in oocytes injected with the BPR mutant, explaining the lack of effect of this mutant in the oocyte, and can even finally interact with the mutant itself. We did not add these results to the manuscript as they are rather marginal in respect of the main issue of the paper. Moreover, since the mutant does not impair PP2A activity towards Arpp19 in *Xenopus* oocytes, it does not add any new information to our conclusions.

17. Reviewer: p. 22, discussion: it is mentioned that "...PP2A-B55delta...is inhibited by Arpp19 phosphorylated at Ser67, a property not shared by the three other PP2A-B55 holoenzymes." This reviewer could not find the evidence for this in the listed references. Could the authors clarify this point.

Authors: We agree with the reviewer that this statement deserves to be modulated. Mochida *et al.*¹ and Castilho *et al.*¹⁹ show that the phosphatase regulating the Cdk1 substrates is PP2A-B55 δ . They exclude other phosphatases (PP1, PP4, PP5 and PP6) and the isoform PP2A-B56 (B56 ϵ and γ) and they

point B55 δ based on the effects of specific depletions of B55 δ . Castilho *et al.*¹⁹ show that the control of M-phase is ensured by the Greatwall inhibition of PP2A-B55 δ , and not by other B55 isoforms. Indeed, the specific depletion of B55 δ (which does not affect the other B55 isoforms) is sufficient to rescue the effects of Greatwall depletion (Figures 9 and 10¹⁹). In 2010, it was discovered that the inhibition of PP2A-B55 δ by Greatwall is specifically mediated by S67-phosphorylated Arpp19^{20, 21}. Mochida *et al.*²¹ show that Arpp19/Ensa is highly specific for the particular species of PP2A that contains the B55 δ subunit and does not bind to other forms of PP2A containing the B56 ϵ , B56 γ or B''/PR48 regulatory subunits (Fig. 2b in²¹). As the specific PP2A-B55 δ isoform is targeted by Greatwall, as Greatwall action is solely mediated by Arpp19 and as Arpp19 specifically binds PP2A-B55 δ but not other PP2A isoforms, we concluded that “PP2A-B55 δ is the key phosphatase isoform that acts on Cdk1 substrates and is inhibited by Arpp19 phosphorylated at S67, a property not shared by the three other *Xenopus* PP2A-B55 holoenzymes”. In order to be more accurate and closer from the direct data, we changed the text (page 14, lines 422-425), now: “PP2A-B55 δ is the key phosphatase isoform that acts on Cdk1 substrates, a property not shared by the other *Xenopus* PP2A-B55 holoenzymes (Mochida *et al.*, 2009; Castilho *et al.*, 2009) and is inhibited by Arpp19 phosphorylated at S67 (Mochida *et al.*, 2010).”

18. Reviewer: p. 22, discussion: *Wee1/Myt1* are listed as PKA substrates. The reference listed investigates this in mouse and shows phosphorylation of *Wee1B*. To this reviewer's knowledge *Wee1B* should not be expressed in *Xenopus* prophase I arrested oocytes. To my knowledge, for *Myt1* it is unclear if it is a PKA target. Could the authors comment on this?

Authors: We agree with the reviewer and were not clear enough in our wording. *Wee1b* is clearly phosphorylated by PKA in mouse oocytes²² and *Wee1* is not expressed in *Xenopus* prophase oocytes, when PKA is active²³. Concerning PKA substrates other than Arpp19 in *Xenopus* oocytes, we intended to mention *Cdc25* and the possibility that, similarly to *Wee1b* in mouse, *Myt1* could be one of these substrates. We have re-written the sentence in a more appropriate way (page 14, lines 434-436), now: “As such, PKA certainly targets substrates other than Arpp19, important to keep oocytes arrested in prophase, such as *Cdc25* (Duckworth *et al.*, 2002) or, similarly to *Wee1b* in mouse {Han *et al.*, 2005}, *Myt1*.”

19. Reviewer: p. 23, discussion: the authors speculate that pS109 Arpp19 might be a PP1 inhibitor. This has been tested by Mochida *et al.*, 2010 and it seems not to be the case.

Authors: Indeed, Supplementary Fig. 1c of the article by Mochida *et al.*²¹ illustrates an *in vitro* assay where the catalytic subunit of PP1 is able to dephosphorylate phosphorylase-a despite the presence of various phosphorylated forms of Arpp19, including the S109-phosphorylated one. However, we cannot exclude that *in vivo*, S109-phosphorylated Arpp19 controls PP1 activity towards specific relevant substrates and acts through a regulatory subunit of PP1 and not directly on the catalytic subunit. We changed the text to be more precise (page 15, lines 459-462).

20. Reviewer: Overall, it seems that PP2A-B56 (as mentioned by authors) or other PPTases significantly contribute to pS109 Arpp19 dephosphorylation. Could the authors comment on the following points:

Authors, general comment on point #20: In the experiment 1, the MonoQ fractionation produced two distinct groups of fractions containing S109-phosphatase activity, one enriched in B55 and the other one in B56 (although the latter also contains low levels of B55). Two additional experiments showed that the B56 content in the active fractions is variable whereas B55 is constantly present in those fractions. Therefore, as stated in the “Discussion”, while PP2A-B55 is the main phosphatase responsible for S109 dephosphorylation of Arpp19, PP2A-B56 can also contribute, at least in extracts. Moreover, although the B subunit specifically and markedly facilitates dephosphorylation of the substrates by PP2A, the dimer PP2A-A/C can exhibit a low activity *in vitro*²⁴ and could account for some S109-phosphatase activity found in fractions of the isolation procedure. To resolve the ambiguity

between B55 and B56, we carried out *in vivo* experiments. Inhibiting PP2A-B55 δ using S67-phosphorylated Arpp19 in intact oocytes fully abolishes S109 dephosphorylation of endogenous Arpp19 induced by progesterone and even increases its phosphorylation level. These experiments unequivocally establish that *in vivo*, PP2A-B55 is the physiological phosphatase that dephosphorylates Arpp19 at S109.

20.1. *The profile of S109 phosphatase activity (mainly in fractions 5-8) in Fig.3B does not fit completely to the elution profile of PP2A-B55 δ (mainly in fraction 4-6).*

Authors: We agree with the reviewer and had commented this observation in our manuscript. Previous Fig. 3b (now Fig. 3c) shows that fraction 7 contains a mix of B55 δ and B56, and fraction 8 contains mainly B56. This argues that *in vitro*, in extracts, PP2A-B56 can contribute to Arpp19 dephosphorylation at S109, as mentioned (page 8, lines 229-230; page 13, lines 397-398).

20.2. *Fig4A: there is much less B55 δ in fraction 12 than fraction 10, however for the S109 phosphatase activity the opposite is true. This is similar for Supplementary Figure 3B lanes 7-9.*

Authors: The inputs loaded on both chromatography columns (PhenySuperose in Fig. 4a and Superose 12 in Supplementary Fig. 2d) correspond to fractions of the previous columns (Q5 MonoQ for the first one and PS11 Phenyl-Superose for the second one) that did not contain any B56 subunit. Moreover, B56 is not detected by mass spectrometry in these fractions (as well as any other phosphatases except PP2A-B55). Therefore, we consider that the B55 isoform is indeed the only phosphatase dephosphorylating Arpp19 in these fractions, to the exclusion of any other, and that these slight differences between the S109-phosphatase activity and the western blot profiles are not significant.

20.3. *Fig 5B: the activity seems to correlate exclusively with B56 ϵ (levels also high in MS analysis of the corresponding fraction).*

Authors: In Fig. 5b, the active fractions contain B56 but also some low levels of B55. In order to discriminate which of the two isoforms is physiologically active, we carried out *in vivo* experiments that removed any ambiguity (former Fig. 7, now Fig. 8c-e).

20.4. *Fig. 6C+D: The μ Cys and tpS67 precipitate similar amounts of B55, however the pS109 phosphatase activity is much higher with the μ Cys, which suggests that another PP2A holoenzyme contributes significantly. Can the authors provide a B56 blot for the experiment shown in Fig. 6C?*

Authors: To answer several requests of the reviewer, we now provide a new figure (Fig. 7e-h and Supplementary Fig. 3c), which includes one representative experiment (same as the former one since we kept the samples) as well as statistics. These new Fig. 7 and Supp Fig. 3c answer points 12 (depletion of PP2A-C), 20.4 (we provide a B56 blot) and 4 (statistics). Supp Fig. 3c shows that, as expected, B56 ϵ is present on μ Cys beads but undetectable on tpS67 beads. Under both conditions, the supernatants depleted either in all PP2A isoforms (μ Cys beads) or specifically in PP2A-B55 δ (tpS67 beads) no longer contain phosphatase activity (Fig. 7e-h).

20.5. *The appearance of PP2A-B56 in the three different MS experiments is somewhat inconsistent. Can the authors explain where it went in experiment 3, where it is still present in the input.*

Authors: In experiment 3, B56 was present in the input (PS11, meaning fraction 11 from the previous PhenylSuperose column) loaded on the last chromatographic step, a Superose column. As in all experiments, we did not analyze by western blot all the fractions (24 in this case) but only fractions 1 to 13 surrounding the S109-phosphatase active fractions (fractions 7 and 8). B56 was not detectable in these fractions. We analyzed by mass spectrometry the fractions containing the peak of S109-phosphatase activity (fractions 7 and 8) and the immediate flanking fractions (fractions 5, 6, 9 and 10) where B56 was not detected as well. Hence, B56 was probably eluted in fractions 13 to 24 and was not implied in S109-phosphatase activity.

References

1. Mochida S, Ikeo S, Gannon J, Hunt T. Regulated activity of PP2A-B55 delta is crucial for controlling entry into and exit from mitosis in Xenopus egg extracts. *EMBO J* **28**, 2777-2785 (2009).

2. Hara M, Abe Y, Tanaka T, Yamamoto T, Okumura E, Kishimoto T. Greatwall kinase and cyclin B-Cdk1 are both critical constituents of M-phase-promoting factor. *Nature communications* **3**, 1059 (2012).
3. Lorca T, *et al.* Constant regulation of both the MPF amplification loop and the Greatwall-PP2A pathway is required for metaphase II arrest and correct entry into the first embryonic cell cycle. *J Cell Sci* **123**, 2281-2291 (2010).
4. Jeong AL, Yang Y. PP2A function toward mitotic kinases and substrates during the cell cycle. *BMB Rep* **46**, 289-294 (2013).
5. Pal G, Paraz MT, Kellogg DR. Regulation of Mih1/Cdc25 by protein phosphatase 2A and casein kinase 1. *J Cell Biol* **180**, 931-945 (2008).
6. Lucena R, Alcaide-Gavilan M, Anastasia SD, Kellogg DR. Wee1 and Cdc25 are controlled by conserved PP2A-dependent mechanisms in fission yeast. *Cell Cycle* **16**, 428-435 (2017).
7. Rata S, *et al.* Two Interlinked Bistable Switches Govern Mitotic Control in Mammalian Cells. *Curr Biol* **28**, 3824-3832 e3826 (2018).
8. Yu J, Zhao Y, Li Z, Galas S, Goldberg ML. Greatwall kinase participates in the Cdc2 autoregulatory loop in *Xenopus* egg extracts. *Mol Cell* **22**, 83-91 (2006).
9. Dupre A, Buffin E, Roustan C, Nairn AC, Jessus C, Haccard O. The phosphorylation of ARPP19 by Greatwall renders the auto-amplification of MPF independently of PKA in *Xenopus* oocytes. *J Cell Sci* **126**, 3916-3926 (2013).
10. Dupre AI, Haccard O, Jessus C. The greatwall kinase is dominant over PKA in controlling the antagonistic function of ARPP19 in *Xenopus* oocytes. *Cell Cycle* **16**, 1440-1452 (2017).
11. Hegarat N, *et al.* Cyclin A triggers Mitosis either via the Greatwall kinase pathway or Cyclin B. *EMBO J*, e104419 (2020).
12. Woodland HR, Pestell RQ. Determination of the nucleoside triphosphate contents of eggs and oocytes of *Xenopus laevis*. *Biochem J* **127**, 597-605 (1972).
13. Newmeyer DD, Lucocq JM, Burglin TR, De Robertis EM. Assembly in vitro of nuclei active in nuclear protein transport: ATP is required for nucleoplasmic accumulation. *EMBO J* **5**, 501-510 (1986).
14. Murray AW, Kirschner MW. Cyclin synthesis drives the early embryonic cell cycle. *Nature* **339**, 275-280 (1989).
15. Robert J, Wolff J, Jijakli H, Graf JD, Karch F, Kobel HR. Developmental expression of the creatine kinase isozyme system of *Xenopus*: maternally derived CK-IV isoform persists far beyond the degradation of its maternal mRNA and into the zygotic expression period. *Development* **108**, 507-514 (1990).
16. Williams BC, *et al.* Greatwall-phosphorylated Endosulfine is both an inhibitor and a substrate of PP2A-B55 heterotrimers. *eLife* **3**, e01695 (2014).
17. Dupre A, Daldello EM, Nairn AC, Jessus C, Haccard O. Phosphorylation of ARPP19 by protein kinase A prevents meiosis resumption in *Xenopus* oocytes. *Nature communications* **5**, 3318 (2014).
18. Cundell MJ, *et al.* A PP2A-B55 recognition signal controls substrate dephosphorylation kinetics during mitotic exit. *J Cell Biol* **214**, 539-554 (2016).
19. Castilho PV, Williams BC, Mochida S, Zhao Y, Goldberg ML. The M phase kinase Greatwall (Gwl) promotes inactivation of PP2A/B55delta, a phosphatase directed against CDK phosphosites. *Mol Biol Cell* **20**, 4777-4789 (2009).
20. Gharbi-Ayachi A, *et al.* The Substrate of Greatwall Kinase, Arpp19, Controls Mitosis by Inhibiting Protein Phosphatase 2A. *Science* **330**, 1673-1677 (2010).
21. Mochida S, Maslen SL, Skehel M, Hunt T. Greatwall phosphorylates an inhibitor of protein phosphatase 2A that is essential for mitosis. *Science* **330**, 1670-1673 (2010).
22. Han SJ, Chen R, Paronetto MP, Conti M. Wee1B is an oocyte-specific kinase involved in the control of meiotic arrest in the mouse. *Curr Biol* **15**, 1670-1676 (2005).
23. Murakami MS, VandeWoude GF. Analysis of the early embryonic cell cycles of *Xenopus*; regulation of cell cycle length by *Xe-wee1* and *Mos*. *Development* **125**, 237-248 (1998).
24. Xu Y, Chen Y, Zhang P, Jeffrey PD, Shi Y. Structure of a protein phosphatase 2A holoenzyme: insights into B55-mediated Tau dephosphorylation. *Mol Cell* **31**, 873-885 (2008).

Changes in Figures

Former Figure 1 (actual Figure 1):

Panel a: now Fig.1a and 1b. New experiments including the control of ATP level in extracts (Request #5 of reviewer 2) and statistics (Request #4 of reviewer 2).

Panel b: now Fig. 1c and 1d. New experiments including the control of ATP level in extracts (Request #5 of reviewer 2) and statistics (Request #4 of reviewer 2).

Panel c: now Fig. 1e and 1f. New experiments including statistics (Request #4 of reviewer 2).

Panel d: now Fig. 8a and 8b, for reasons of scientific logic in the presentation of results.

Former Figure 2 (actual Figure 2):

Panel b: now Fig.2b and 2c. New panel c presents statistics from several experiments identical to the representative one illustrated in panel b (Request #4 of reviewer 2).

Panel c: now Fig.2d and 2e. New panel e presents statistics from several experiments identical to the representative one illustrated in panel d (Request #4 of reviewer 2).

Former Figure 3 (actual Figure 3):

Insertion of a new panel, panel a: the flow chart in former Supplementary Figure 2 was moved to this main figure (Request #2 of reviewer 1).

Insertion of a new Figure, Figure 6:

This new figure illustrates new experiments using purified PP1 and B55 δ (Request #1 of reviewer 1).

Former Figure 6 (now Figure 7):

Panels a and b are now Supplementary Figure 3a and 3b. New panels a, b, c and d of new Fig. 7 illustrate new experiments showing the kinetics of S109 and S67 dephosphorylation of doubly-phosphorylated Arpp19 (Request #10 of reviewer 2) and statistics (Request #4 of reviewer 2).

Panel c is now Supplementary Figure 3c, and corresponds to a new western blot with additional antibodies (Requests #12 and 20.4 of reviewer 2).

Panel d: was removed. Replaced by new panels e to h in new Fig. 7, illustrating new experiments of PP2A depletion using either μ Cys-beads or tpS67-beads (Request #12 of reviewer 2) and statistics (Request #4 of reviewer 2).

Former Figure 7 (now Figure 8):

Introduction of new panels a and b corresponding to former Fig. 1d.

Panel a: now Fig. 8c.

Panel b: now Fig. 8d and 8e. Change in the presentation of the quantification for reasons of consistency along the article.

Former Figure 8 (now Figure 9):

Panel a: now Fig. 9a and 9b. Change in the presentation of the quantification for reasons of consistency along the article.

Panel b: now Supplementary Fig. 5a and 5b for reasons of space constraints.

Panel c: now Fig. 9c

Panel d: now Fig. 9d. Change in the presentation of the quantification for reasons of consistency along the article.

Panel e: now Fig. 9e and 9f, providing quantifications and statistics (Request #4 of reviewer 2).

New Supplementary Figure 1:

New experiments answering request #11 of reviewer 2.

Former Supplementary Figure 1:

Now Supplementary Fig. 4, due to its occurrence in the main text.

Former Supplementary Figure 2:

Panel a: now Supp Fig. 2a and 2b, the latter presenting statistics from several experiments identical to the representative one illustrated in panel a (Request #4 of reviewer 2).

Panel b: now Fig. 3a (Request #2 of reviewer 1).

Former Supplementary Figure 3:

Panels a and b are now Supplementary Fig. 2c and 2d.

New Supplementary Figure 3:

Panels a and b corresponds to former Fig. 6a and 6b.

Panel c: corresponds to a new western blot with additional antibodies (Requests #12 and 20.4 of reviewer 2), equivalent to former Fig. 6c.

New Supplementary Figure 4:

Corresponds to former Supplementary Fig. 1.

New Supplementary Figure 5:

Corresponds to former Fig. 8b.

REVIEWERS' COMMENTS

Reviewer #1 (Remarks to the Author):

The manuscript has been carefully revised and my concerns have been mostly addressed. In particular, the presentation of the impressive chromatography experiments has been significantly improved. I have just one minor comment: in Figure 6d, the authors compare dephosphorylation of pS109-GST-Arpp19 by PP2A-B55 and PP1. The total levels of GST-Arpp19 seem to drop during the course of the experiment when PP1 is used as a phosphatase, but not with PP2A-B55. Is there an explanation for this? Do the authors have a version of this experiment where the total levels of protein are maintained during the experiment? If yes, it would be good to swap the current version for one where the total protein does not change.

Reviewer #2 (Remarks to the Author):

The authors have done a great piece of work to address the points I raised before. The ms provides now important insights into the regulation of meiotic maturation. It should be published as it is.

Point-by-point response to the reviewers' comments

Article title: *The M-phase regulatory phosphatase PP2A-B55 δ opposes protein kinase A on Arpp19 to initiate meiotic division.* Lemonnier et al.

Reviewer #1

Remarks to the Author:

The manuscript has been carefully revised and my concerns have been mostly addressed. In particular, the presentation of the impressive chromatography experiments has been significantly improved. I have just one minor comment: in Figure 6d, the authors compare dephosphorylation of pS109-GST-Arpp19 by PP2A-B55 and PP1. The total levels of GST-Arpp19 seem to drop during the course of the experiment when PP1 is used as a phosphatase, but not with PP2A-B55. Is there an explanation for this? Do the authors have a version of this experiment where the total levels of protein are maintained during the experiment? If yes, it would be good to swap the current version for one where the total protein does not change.

Authors' response:

We thank the reviewer for his/her positive feedback on the revised version of our article and his/her suggestion for improvement of Figure 6d. Indeed, the total levels of GST-Arpp19 seem to drop during the time-course of the experiment with PP1. The approach used to monitor Arpp19 dephosphorylation generates slight variations in substrate levels, as can be seen by eye on western blots. To avoid any bias in the conclusions of the experiments due to these variations, all experiments were quantified and normalized as described in the "Methods" section. Figure 6e illustrates the quantified measures of 5 experiments. Figure 6d is just a representative experiment, as mentioned in the figure legend and in the "Methods" section.

As requested by the referee, a new experiment is now illustrated in Figure 6d. This new experiment was already included in the quantifications reported in Figure 6e. Please see below the former experiment and the new one. According to a visual assessment of this new experiment, the total levels of ⁸⁵S-Arpp19 are maintained in PP1 and PP2A-B55 δ assays, but slightly vary in the control assay.

Reviewer #2

Remarks to the Author:

The authors have done a great piece of work to address the points I raised before. The ms provides now important insights into the regulation of meiotic maturation. It should be published as it is.

Authors' response:

We thank the reviewer for his/her positive assessment.

Changes

The text has been changed to comply with the editorial policies and formatting requirements of Nature Communications. Modifications are highlighted in yellow. The change of panel 6d did not impose any modification of the text since it represents a new experience identical to the previous one.

Figure 6:

Panel "d" of Figure 6 has been replaced by a new experiment at the request of reviewer 1.